# *Escherichia coli* possessing the dihydroxyacetone phosphate shunt utilize 5′-deoxynucleosides for growth

Katherine A. Huening,[1] Joshua T. Groves,[1] John A. Wildenthal,[1] F. Robert Tabita,[1] Justin A. North[1]

**ABSTRACT** All organisms utilize S-adenosyl-L-methionine (SAM) as a key co-substrate for the methylation of biological molecules, the synthesis of polyamines, and radical SAM reactions. When these processes occur, 5′-deoxy-nucleosides are formed as byproducts such as S-adenosyl-L-homocysteine, 5′-methylthioadenosine (MTA), and 5′-deoxy-adenosine (5dAdo). A prevalent pathway found in bacteria for the metabolism of MTA and 5dAdo is the dihydroxyacetone phosphate (DHAP) shunt, which converts these compounds into dihydroxyacetone phosphate and 2-methylthioacetaldehyde or acetaldehyde, respectively. Previous work in other organisms has shown that the DHAP shunt can enable methionine synthesis from MTA or serve as an MTA and 5dAdo detoxification pathway. Rather, the DHAP shunt in *Escherichia coli* ATCC 25922, when introduced into *E. coli* K-12, enables the use of 5dAdo and MTA as a carbon source for growth. When MTA is the substrate, the sulfur component is not significantly recycled back to methionine but rather accumulates as 2-methylthioethanol, which is slowly oxidized non-enzymatically under aerobic conditions. The DHAP shunt in ATCC 25922 is active under oxic and anoxic conditions. Growth using 5-deoxy-D-ribose was observed during aerobic respiration and anaerobic respiration with Trimethylamine N-oxide (TMAO), but not during fermentation or respiration with nitrate. This suggests the DHAP shunt may only be relevant for extraintestinal pathogenic *E. coli* lineages with the DHAP shunt that inhabit oxic or TMAO-rich extraintestinal environments. This reveals a heretofore overlooked role of the DHAP shunt in carbon and energy metabolism from ubiquitous SAM utilization byproducts and suggests a similar role may occur in other pathogenic and non-pathogenic bacteria with the DHAP shunt.

**IMPORTANCE** The acquisition and utilization of organic compounds that serve as growth substrates are essential for *Escherichia coli* to grow and multiply. Ubiquitous enzymatic reactions involving S-adenosyl-L-methionine as a co-substrate by all organisms result in the formation of the 5′-deoxy-nucleoside byproducts, 5′-methylthioadenosine and 5′-deoxyadenosine. All *E. coli* possess a conserved nucleosidase that cleaves these 5′-deoxy-nucleosides into 5-deoxy-pentose sugars for adenine salvage. The DHAP shunt pathway is found in some extraintestinal pathogenic *E. coli*, but its function in *E. coli* possessing it has remained unknown. This study reveals that the DHAP shunt enables the utilization of 5′-deoxy-nucleosides and 5-deoxy-pentose sugars as growth substrates in *E. coli* strains with the pathway during aerobic respiration and anaerobic respiration with TMAO, but not fermentative growth. This provides an insight into the diversity of sugar compounds accessible by *E. coli* with the DHAP shunt and suggests that the DHAP shunt is primarily relevant in oxic or TMAO-rich extraintestinal environments.

**KEYWORDS** extraintestinal pathogenic *E. coli*, nucleotide metabolism, carbon metabolism, sulfur

Address correspondence to Justin A. North, north.62@osu.edu.

F. Robert Tabita passed away on 5 January 2021 during the preparation of this manuscript.

The authors declare no conflict of interest.

See the funding table on p. 16.

Infection caused by *Escherichia coli* is a major global health concern with significant health and economic impacts (1–3). Extraintestinal pathogenic *E. coli* (ExPEC) strains can cause infection in humans and animals in niches outside of the gut (2, 4). These include uropathogenic *E. coli* (UPEC), a major cause of urinary tract infections (UTIs), along with strains that cause blood, prostate, and mammary infections, as well as neonatal meningitis (1, 5). ExPEC strains are diverse, harboring numerous horizontally acquired pathogenicity, fitness, and other genes located within genomic islands (6, 7). While the function of many of these genes has been identified, such as involvement in adhesion, iron uptake, and toxin production (3, 4, 7), many others remain unknown in their function or physiological role (8–10). Previously using comparative genomics, we observed that 42% of *E. coli* isolates in sequenced UTI and blood infection isolate databases possess a conserved gene cluster known as the dihydroxyacetone phosphate (DHAP) shunt for the salvage of 5′-deoxy-nucleosides and 5-deoxy-pentose sugars (Fig. 1B and C) (11). For example, the DHAP shunt is absent in commonly used model UPEC strains such as 536, UTI89, and CFT073 but is observed in ATCC 25922, a CFT073 phylogenetic relative based on genetic marker and virulence gene analysis (3, 12), F11, and members of the globally disseminated ExPEC ST131 lineage (e.g., EC958 and NA114) (11). In contrast, <0.1% of analyzed pathogenic intestinal *E. coli* isolates (1 of 1,376) contained the DHAP shunt gene cluster.

The *E. coli* variation of the DHAP shunt is composed of a nucleosidase (Pfs, known as MtnN in other organisms), a kinase (MtnK), an isomerase (MtnA), and an aldolase (Ald2) that function to convert 5′-methylthioadenosine (MTA) and 5′-deoxyadenosine (5dAdo) into adenine, DHAP, and an aldehyde (Fig. 1B and C). The other known DHAP shunt variation, which does not appear in *E. coli*, employs a bifunctional nucleosidase/phosphorylase (MtnP) that replaces the nucleosidase and kinase (11, 16, 17). All *E. coli* possess the Pfs nucleosidase for purine salvage and the disposal of MTA and 5dAdo to prevent inhibitory buildup (Fig. 1B) (18–20). In ExPEC strains with the DHAP shunt, the characteristic *mtnK, mtnA,* and *ald2* genes appear to be located as a single conserved gene cluster at the end of the tRNA-*leuX* genomic island (21). For environmental freshwater and soil bacteria like *Rhodospirillum rubrum* and *Rhodopseudomonas palustris*, the DHAP shunt was shown to function physiologically as a sulfur salvage pathway. In these organisms, MTA was metabolized to 2-methylthioacetaldehyde for the synthesis of the amino acid methionine (16, 22–24) (Fig. S1D). In the insect pathogen, *Bacillus thuringiensis*, which also possesses the universal methionine salvage pathway for MTA recycling to methionine (Fig. S1B), the DHAP shunt was observed to serve as a detoxification pathway for 5dAdo and 5-deoxy-D-ribose (5dR) (17).

MTA, 5dAdo, and *S*-adenosyl-L-homocysteine (SAH) are metabolic byproducts formed by all organisms via enzymatic reactions using *S*-adenosyl-L-methionine (SAM; Fig. 1). They are competitive inhibitors of SAM-utilizing enzymes, and concentrations above 1 mM are inhibitory to *E. coli* growth (15, 19, 20). Therefore, cells must either convert them to non-inhibitory compounds and/or excrete them outside of the cell (18). In *E. coli*, SAH is initially cleaved by the MTA/5dAdo/SAH nucleosidase (Pfs) to produce adenine and *S*-ribosyl-L-homocysteine, which can continue in the methionine cycle to regenerate methionine and SAM (Fig. 1A) (15). In *E. coli* without the DHAP shunt (e.g., K-12 and Nissle), adenine is cleaved from MTA and 5dAdo by Pfs, but the resulting 5-deoxy-pentoses, 5-methylthio-D-ribose (MTR), and 5-deoxy-D-ribose, respectively, are excreted from the cell (Fig. 1B) (18, 19, 25).

Previously we reported that the DHAP shunt was active in *E. coli* ATCC 25922, an attenuated ExPEC strain originally isolated from a clinical patient in Seattle, Washington (26, 27), for both MTA and 5dAdo metabolism. While ExPEC strains like 536 and CFT073 are highly virulent in mouse models of septicemia, ATCC 25922 exhibits moderate virulence (26, 28) and can also establish infections in the lungs, liver, heart, spleen, and kidneys of mice (29). However, the physiological role of the DHAP shunt in *E. coli* that possessed it was unknown. Here, we report that *E. coli* ATCC 25922 uses the DHAP shunt for growth with externally acquired 5′-deoxy-nucleosides and 5-deoxy-pentose sugars

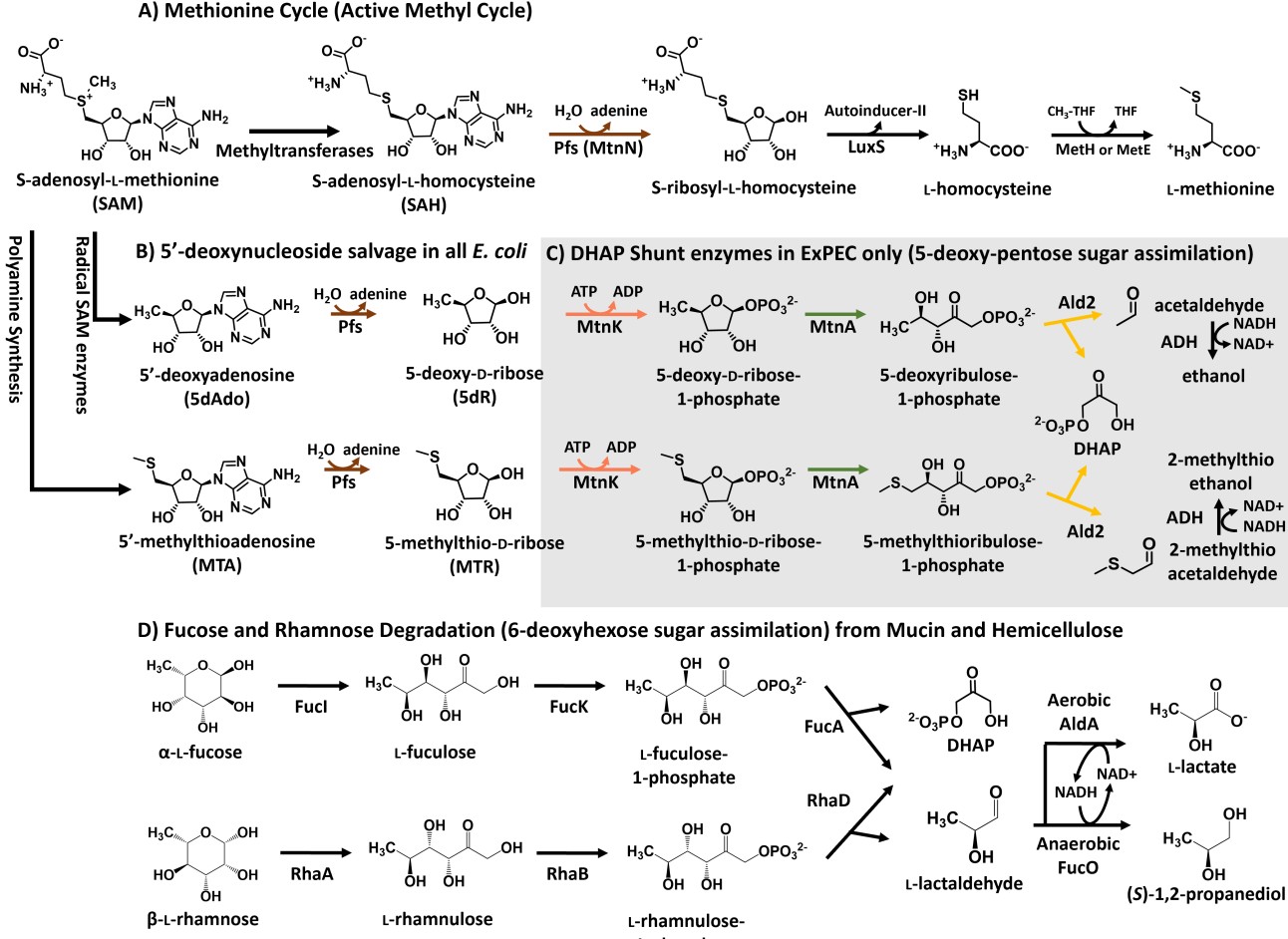

**FIG 1** Salvage of S-adenosyl-L-methionine (SAM) utilization byproducts, 5-deoxy-pentose sugars, and 6-deoxy-hexose sugars in *E. coli*. (A) *S*-adenosyl-L-homo-cysteine (SAH), produced by methyltransferases, is recycled by the methionine cycle (a.k.a. active methyl cycle) (13, 14). (B and C) The *E. coli* variation of the DHAP shunt. (B) For adenine salvage and detoxification of 5′-methylthioadenosine (MTA) and 5′-deoxyadenosine (5dAdo), all *E. coli* possess the multifunctional SAH/MTA/5dAdo nucleosidase (Pfs; a.k.a. MtnN) (15). (C) ExPEC strains possess the Pfs nucleosidase (B) and remainder of the DHAP shunt genes, which together compose a dual-purpose pathway for the conversion of 5′-deoxy-nucleosides in the form of MTA and 5dAdo or corresponding 5-deoxy-pentoses in the form of 5-deoxy-D-ribose and 5-methylthioribose, respectively, into the central carbon metabolite DHAP and acetaldehyde or (2-methylthio)acetaldehyde (11). (D) *E. coli* analogously metabolize 6-deoxy-hexose sugars in the form of L-fucose and L-rhamnose to DHAP and L-lactaldehyde. During anaerobic growth, L-lactaldehyde is primarily reduced to (*S*)-1,2-propanediol as a terminal product, whereas during aerobic growth, it is primarily oxidized to L-lactate for carbon assimilation as pyruvate. Enzymes: Pfs (MtnN), SAH/MTA/5dAdo nucleosidase, E.C. 3.2.2.9; LuxS, *S*-ribosyl-L-homocysteine lyase, E.C. 4.4.1.21; MetH, methionine synthase, E.C. 2.1.1.13; MetE, methionine synthase, E.C. 2.1.1.4; MtnK, 5-methylthioribose/5-deoxyribose kinase, E.C. 2.7.1.100; MtnA, 5-methylthioribose-1-phosphate/5-deoxyribose-1-phosphate isomerase, E.C. 5.3.1.23; Ald2, 5-methylthioribulose-1-phosphate/5-deoxyribulose-1-phosphate aldolase, E.C. 4.1.2.62; ADH, alcohol dehydrogenase, E.C. 1.1.1.1 ; FucI, L-fucose isomerase, E.C. 5.3.1.25; FucK, L-fuculose kinase, E.C. 2.7.1.51; FucA, L-fuculose-1-phosphate aldolase, E.C. 4.1.2.17; RhaA, L-rhamnose isomerase, E.C. 5.3.1.14; RhaB, L-rhamnulose kinase, E.C. 2.7.1.5; RhaD, L-rhamnulose-1-phosphate aldolase, E.C. 4.1.2.19; FucO, (*S*)−1,2-propanediol oxidoreductase, E.C. 1.1.1.77; and AldA, L-lactaldehyde dehydrogenase, E.C. 1.2.1.22.

as a carbon source (C-source), analogous to 6-deoxy-hexose sugar metabolism by the fucose and rhamnose catabolic pathways of *E. coli* (Fig. 1B through D). Furthermore, the transfer of the DHAP shunt gene cluster to reference strain K-12 is sufficient to enable aerobic growth with 5dAdo and 5dR. This distinguishes *E. coli*'s use of the DHAP shunt pathway from other environmental organisms previously described, which employ it for sulfur salvage (Fig. S1D) (11, 16, 22) or 5-deoxy-pentose detoxification (17).

## MATERIALS AND METHODS

### Fine chemicals

All chemicals unless otherwise stated were from Millipore-Sigma. All gases for growth studies were of >99.99% purity from Praxair. All isotopically labeled standards were synthesized exactly as previously described (11, 16).

### Strains and growth conditions

#### Bacterial strains

*E. coli* clinical isolate ATCC 25922 (Seattle 1946) (12, 26, 27) was used as a representative strain containing the DHAP shunt genes. BW25113, the parent strain of the Keio Collection, which is derived from *E. coli* K-12 (30), was used as a control strain naturally lacking the DHAP shunt. Deletion of gene cluster *mtnK-mtnA-ald2* in ATCC 25922 (strain ΔK2; Δ*mtnK* Δ*mtnA* Δ*ald2*) and the *pfs* gene was performed using the λ-red system (31) as previously described (11). Primers used for PCR amplification from pKD4 for each gene deletion are listed in Table S1. *E. coli* Stellar (Takara Biosciences) was used for cloning and plasmid storage. All transformations were performed by electroporation.

#### General growth

Cells were either grown in Lysogeny Broth (LB) or modified M9 media as previously described (11), supplemented with 1 mg/L nicotinic acid (ATCC 25922 is an auxotroph), 40 mM sodium nitrate unless otherwise indicated, 1 mM ammonium sulfate unless otherwise indicated, and either glucose, 5dR, MTA, or 5dAdo at indicated concentrations. Aerobic and anaerobic cultures were incubated at 37°C with shaking at 250 rpm. For liquid growth experiments, ATCC 25922 was first grown overnight aerobically in liquid LB cultures, then washed three times in carbon-free or sulfur-free M9 media prior to inoculation in M9 media to an initial optical density at 600 nm ($OD_{600nm}$) of ~0.05. Anaerobic manipulations were performed in an anaerobic chamber with 5%/95% $H_2/N_2$ atmosphere (Coy Laboratories).

### Growth inhibition studies

*E. coli* strains were grown aerobically in M9 with 25 mM glucose to the mid-exponential phase, washed once with M9 glucose media, and used to inoculate 2 mL of M9 glucose media supplemented with MTA, 5dAdo, or 5dR to an initial $OD_{600nm}$ of ~0.2. Cells were grown aerobically for 18 hours to allow each culture to reach its final maximum optical density. $LD_{50}$ values were determined by weighted nonlinear parametric fit (MatLab, Mathworks) to the Hill equation using the average and standard deviation of final culture density versus inhibitor concentration for $n = 3$ independent experiments.

### Oxygen limitation

Anaerobic culture tubes containing M9 media with either 25 mM glucose, 5 mM 5dR, or no carbon source were inoculated, stoppered and sealed, and fitted with 18 Ga needles as an inlet and outlet. For 15 min, the culture was sparged with an atmosphere of defined oxygen concentration made by mixing purified air together with $N_2$ via a gas mixer (Matheson #602; 0 PSIG) at a rate of 100–200 µL/s and 1 atm total pressure. Cultures were incubated at 37°C in a shaking water bath at 120 rpm, and the headspace was purged with the same gas mixture. The flow rate for each gas (air and nitrogen) and coordinately the $O_2$ partial pressure was calculated using the manufacturer's gas flow rate calibration charts (Matheson #602; 0 PSIG).

### Serial plating and growth

Solid agar M9 plates for serial dilution growth studies were made with 1 mM glucose, MTA, 5dAdo, or 5dR; 7.5 g/L Noble agar (Affymetrix); and molecular biology grade water

(Sigma). Cells were initially grown aerobically in 5 mL of M9 media supplemented with 5 mM glucose to the mid-exponential phase, washed four times with carbon-free M9 media, serially diluted onto the plates, and incubated at 37°C for 2–3 days.

## Plasmid construction

An *E. coli* complementation vector, pTETTET, with a tetracycline-inducible promoter was constructed by replacing the *lac* promoter of pBBRsm2-MCS5 (32) with the dual *tet* promoter plus *tetR* and the 70 bp region immediately downstream of *tetR* from the transposon Tn*10* (33, 34). The insertion sequence was synthesized (Table S2) (Genewiz, Azenta Life Sciences), which included flanking NdeI and AseI restriction sites for cloning into pBBRsm2-MCS5 (Fig. S2). In the synthesized fragment, internal NdeI and AseI sites in the *tet* regulon were removed such that only silent mutations were introduced into the *tetR* sequence. The *tet* regulon was amplified using primers Pzt1RBS-AseI-F and Pzt1RBS-NdeI-R (Table S1) and inserted into pBBRsm2-MCS5 to form pTETTET such that the *tetA* promoter was upstream of *lacZa* and the MCS of pBBRsm2-MCS5 (Fig. S2). The *mtnK-mtnA-ald2* gene cluster was then cloned into pTETTET by PCR amplification using primers K2-NdeI-F and K2-SacI-R (Table S1) that introduced flanking NdeI and SacI sites. This resulted in plasmid pK2 for tetracycline-inducible expression of DHAP shunt genes in *E. coli*. The Pfs complementation plasmid, pPfs, was similarly constructed by cloning *pfs* into pTETTET by PCR amplification using primers Pfs-NdeI-F and Pfs-SacI-R (Table S1) that also introduced flanking NdeI and SacI sites.

## Targeted analysis of DHAP shunt metabolites

High-Performance Liquide Chromatography (HPLC) analysis of cellular SAH, MTA, and 5dAdo was performed as previously described (11). Targeted metabolomics of aerobic *E. coli* strains fed with [$^{14}$C-methyl]-5′-methylthioadenosine or [5′-$^{3}$H]–5′-deoxyadenosine was also performed as previously described (11). For 14C-labeled MTA feedings, cells were grown aerobically to the mid-log phase in M9 glucose media, washed three times with sulfur-free M9 glucose media, and resuspended to an OD$_{600nm}$ ~ 5 in sulfur-free M9 glucose media. Cells were fed with a mixture of 15 µM [$^{14}$C-methyl]-5′-methylthioade-nosine (0.1 µCi/100 µL) and 200 µM MTA, and 100 µL aliquots were placed in an open 2 mL conical glass tube, bubbled with purified air, and incubated at 120 rpm at 37°C in a shaking water bath. Samples (100 µL) were collected at indicated time points and flash frozen in LN$_2$. Metabolites were collected by thawing the cells to room tempera-ture, vortexing, centrifuging at 5,000 × $g$ for 1 min, and collecting the spent media supernatant. Metabolites were resolved by C18 reverse phase HPLC with inline liquid scintillation detection (11). [$^{14}$C-methyl]-5′-methylthioadenosine was synthesized from [$^{14}$C-methyl]-*S*-adenosyl-L-methionine (Perkin-Elmer) by acid hydrolysis (11). Similarly, for 5dAdo feedings, anaerobically grown cells were washed with carbon-free M9 media and resuspended to an OD$_{600nm}$ ~ 5 in carbon-free M9 media. Cells were fed with a mixture of 0.4 µM [5′-$^{3}$H]-5′-deoxyadenosine (0.2 µCi/100 µL) and 200 µM 5dAdo and incubated in the shaking water bath as 100 µL aliquots in 2 mL sealed serum vials. Cells were collected at indicated time points by centrifugation at 5,000 × $g$ for 1 min, and metabolites present in the spent media were resolved by ion exclusion chromatography with inline scintillation detection (11).

## Untargeted metabolomics

Untargeted metabolomics were performed at the Ohio State University Campus Chemical Instrument Center. *E. coli* ATCC 25922 was grown in M9 media with 25 mM glucose, 1 mM sulfate, and supplemented with or without 1 mM MTA. Cells were grown aerobically at 37°C with shaking at 250 rpm to the mid-exponential phase and harvested by centrifugation. The culture media were retained, and metabolites were extracted from cells with an equal volume of 50% acetonitrile in water. The combined media and extracted metabolites were lyophilized and resuspended in 50% acetonitrile in water

for LC-MS/MS analysis. Metabolites were resolved using an Agilent ZORBAX SB-Aq C18 reverse phase column (3.0 × 150 mm, 3.5 µm resin) on a Dionex/Thermo UltiMate 3000 HPLC System. Metabolites were eluted on a linear gradient of 5%–20% acetonitrile in water plus 0.1% formic acid. Full mass and MS/MS were acquired by ESI in the negative mode on a Thermo QE M UPLC MS/MS System.

## Cell yield measurements

*E. coli* ATCC 25922 was grown aerobically by shaking at 250 rpm, anaerobically with 100 mM TMAO, or anaerobically with no electron acceptor added in M9 media with 5 mM of each growth substrate. Cells were harvested by centrifugation, spent media were retained, and the cell pellets were lyophilized to dryness to quantify total cell mass. The initial and final concentrations of each substrate before and after growth, respectively, were quantified by C18 reverse phase HPLC using phenylhydrazine derivatization. To each 500 µL sample of spent media, 10 µL of fresh 50 mg/mL phenylhydrazine in 10% glacial acetic acid was added and incubated at 100℃ for 1 hour with occasional vortexing. Metabolites were resolved on a gradient of 17.5%–50% acetonitrile in water with 20 mM ammonium acetate over 20 min with UV detection at 340 nm. Metabolite identities and concentrations were determined based on standard calibration curves of known metabolite standards. Cell yield ($Y_{x/s}$) was calculated as the grams of dry cell weight per gram of substrate consumed.

## RESULTS

### The DHAP shunt in *E. coli* ATCC25922 is not a significant means of sulfur acquisition and leads to accumulation of 2-methylthioethanol and 2-methyl-sulfinylethanol

Seminal studies on the physiological role of the DHAP shunt in environmental photo-synthetic bacteria like *Rhodospirillum rubrum* and *Rhodopseudomonas palustris* revealed that this pathway enabled sulfur salvage from MTA to maintain cellular methionine and cysteine pools (11, 16, 22, 24) (Fig. 1). This occurred through the further metabolism of the terminal DHAP shunt metabolite, 2-methylthioacetaldehyde, into 2-methylthioetha-nol and subsequent cleavage into methanethiol for methionine synthesis (Fig. S1). As a result, *R. rubrum* and *R. palustris* could acquire sulfur and grow using MTA as a sole sulfur source via the DHAP shunt (22, 24). For *E. coli* strain ATCC 25922, initial work showed that the DHAP shunt was active for the metabolism of MTA to 2-methylthioethanol (Fig. 1C) (11). However, it remained unclear as to whether ATCC 25922 could salvage sulfur from MTA for methionine synthesis under aerobic or anaerobic conditions given that it is missing one or more gene homologs for any one of the known methionine salvage pathways present in other organisms (Fig. S1) (11, 16, 22–24, 35–37).

To determine if the *E. coli* DHAP shunt was involved in sulfur salvage from SAM utilization byproducts, we performed growth studies with MTA or 2-methylthioethanol as a sole sulfur source (Fig. 2A). While ATCC 25922 was able to grow using sulfate or methionine as the sole sulfur source, it was completely incapable of growth with MTA or 2-methylthioethanol. This suggests that even if the DHAP shunt enables some sulfur salvage in the *E. coli* that possess it, it is insufficient to support growth unlike in other organisms where the DHAP shunt has been established to function in distinct sulfur salvage pathways for MTA (Fig. S1D) (16, 22, 24). Similarly, we assessed whether the DHAP shunt was involved in recycling endogenously produced SAM utilization byproducts. Quantification of these compounds in the surrounding media enables relative compari-son of SAM byproducts produced between strains and growth conditions (Fig. S3A and B) (11, 19, 38). During aerobic growth, as evidenced by the Pfs deletion, ATCC 25922 produced twofold more MTA than *E. coli* K-12, similar to the levels previously reported in intestinal pathogenic *E. coli* strains due to polyamine synthesis (39), and twofold less SAH (Fig. S3A; *t*-test, $P < 0.05$). The low amounts of 5dAdo produced aerobically are due to the general oxygen sensitivity of radical SAM enzymes (15). For anaerobic

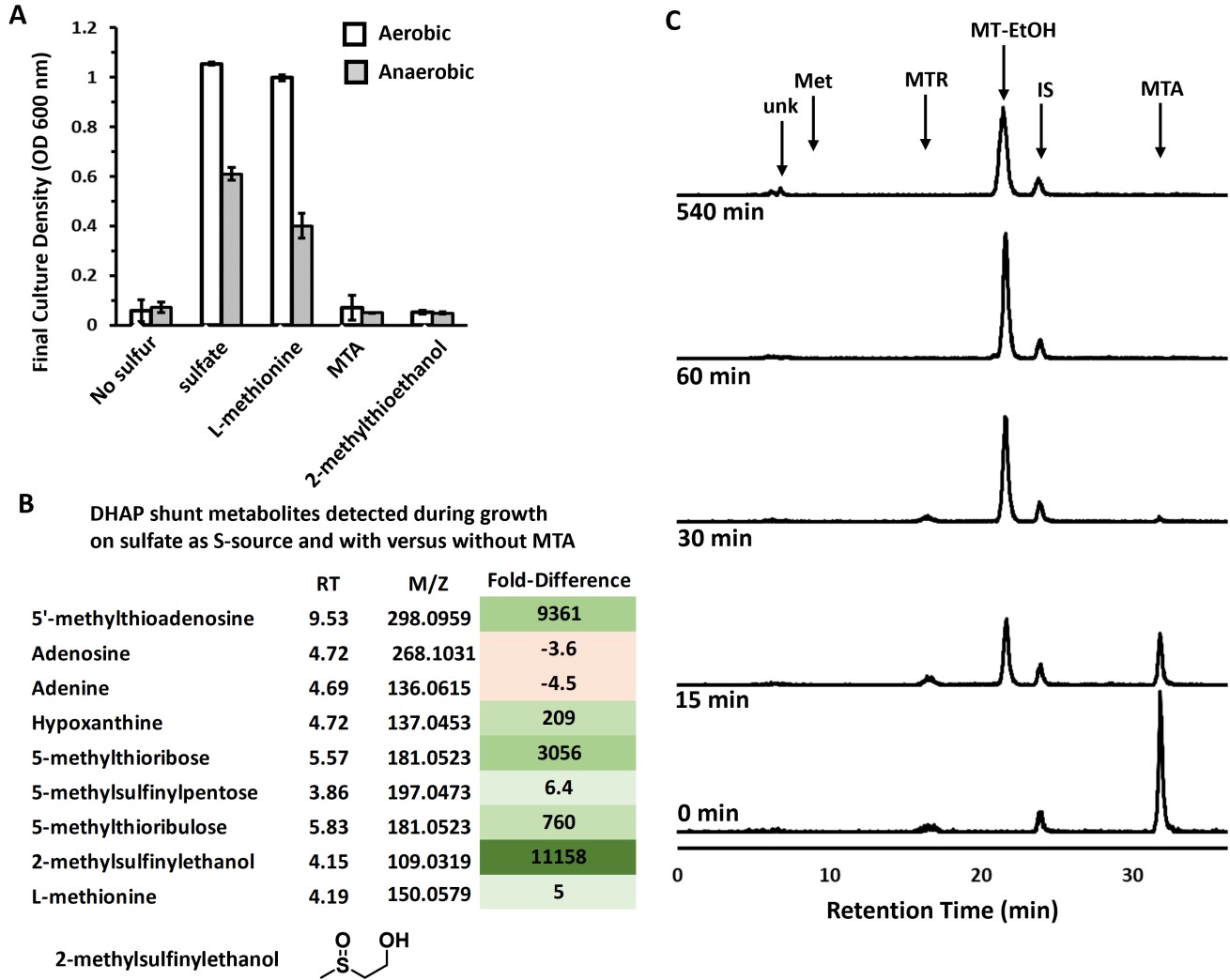

**FIG 2** Sulfur from MTA cannot be salvaged by *E. coli* for growth. (A) ATCC 25922 maximum growth achieved after 24 hours measured by optical density at 600 nm when cultured with 1 mM of the indicated sulfur compound as the sole sulfur source. No further growth was observed after 24 hours. Average and standard deviation error bars are for $n = 3$ independent replicates. (B) Fold difference in the abundance of DHAP shunt-associated metabolites when ATCC 25922 was grown aerobically in the presence of 1 mM sulfate and 1 mM MTA versus grown aerobically in the presence of 1 mM sulfate only. Metabolites were resolved by LC-MS/MS from three independent biological replicates for each growth condition. Values are the average for the three replicates, and the significance of fold change was analyzed by ANOVA with $P < 0.05$. (C) Reverse-phase HPLC quantification of *E. coli* ATCC 25922 metabolites when fed aerobically with [$^{14}$C-methyl] −5′-methylthioadenosine. Unk, unknown (identified as 2-methylsulfinylethanol by LC-MS/MS, $m/z$ = 109.0319), $R_T$ = 6.7 min; Met, methionine, $R_T$ = 8.0 min; MTR, methylthioribose, $R_T$ = 16.7 min; MT-EtOH, 2-methylthioethanol, $R_T$ = 21.7 min; IS, internal standard, $R_T$ = 24.0 min; and MTA, 5′-methylthioadenosine, $R_T$ = 31.8 min.

growth, there was no statistically significant difference in the amounts of SAH, MTA, and 5dAdo produced between ATCC 25922 and K-12 (Fig. S3B; *t*-test, $P > 0.5$). This indicates that *E. coli* strains containing the DHAP shunt do not produce substantially more SAM utilization byproducts than other strains. Coordinately, when the DHAP shunt genes were deleted (ATCC 25922 strain ΔK2; Δ*mtnK*/Δ*mtnA*/Δ*ald2*), there was no defect in growth when sulfate or carbon was limiting compared to the wild-type strain (Fig. S3C and D). Together, this indicates that the DHAP shunt is not significantly involved in the salvage of endogenously produced MTA and 5dAdo, which is in stark contrast to *R. rubrum* and *R. palustris* (11, 16).

To further probe the metabolism of MTA by ATCC 25922, particularly for methionine salvage, we performed untargeted LC-MS/MS metabolomics of ATCC 25922 grown using sulfate supplemented with or without 1 mM MTA and targeted metabolomics of cells fed with isotopically labeled MTA. When grown in the presence of or fed with MTA, there was a large increase in the abundance of DHAP shunt metabolites (Fig. 2B and C; MTA, MTR, and 5-methylthioribulose). Concomitantly, hypoxanthine, which is the deaminated form of adenine, was the only purine that markedly increased in abundance, indicating that excess adenine cleaved from MTA accumulates as hypoxanthine as previously observed in soil and freshwater bacteria (23). Methionine, however, only increased in abundance by approximately fivefold when cells were grown in the presence of MTA (Fig. 2B), and no labeled methionine was detected after feeding with MTA (Fig. 2C; limit of detection of 1 nCi = 15 pmol). This indicates that if methionine is salvaged from MTA, it is at low concentrations (<150 nM). Rather, the change in methionine abundance observed during growth in the presence of MTA is likely due to altered rates of *de novo* methionine synthesis and utilization, given MTA is a competitive inhibitor of SAM-dependent enzymes and a regulator of cell metabolism (15, 23, 40). Thus, the growth and metabolite analyses and the lack of identifiable gene homologs for any of the known methionine salvage pathways for MTA based on the DHAP shunt (Fig. S1D) (11) all support the conclusion that the DHAP shunt is not a significant means of sulfur salvage in *E. coli* that possess it, and any salvage is likely due to unknown, promiscuous processes.

Ultimately for sulfur metabolism from MTA by the DHAP shunt, both the untargeted and targeted metabolomics reveal that 2-methylthioethanol is the primary terminal product (Fig. 2B and C). After 30 min post-feeding with MTA, 100% of the MTA was converted to 2-methylthioethanol, and even after 8 hours post-feeding, only 2-methylthioethanol and a new species ($R_T$ = 6.7 min) were observed. The slow conversion of 2-methylthioethanol ($R_T$ = 21 min) to the new species was found to be non-biological in nature by incubating labeled [2-methyl-$^{14}$C]-methylthioethanol in media aerobically over the same time course as cell feeding and resolving by reverse phase HPLC (Fig. S4B). Coordinately, the untargeted metabolomics identified a compound with mass corresponding to 2-methylsulfinylethanol ($m/z$ = 108.0246 Da, 108.0245 expected, error = 1 ppm) that highly increased in abundance when cells were grown in the presence of MTA (Fig. 2B). The reason 2-methylthioethanol was not observed by untargeted metabolomics is because of removal during the metabolite extraction and lyophilization process for LC-MS/MS (Fig. S4A). We verified that the extraction process does not cause oxidation of 2-methylthioethanol (Fig. S4A), confirming that the formation and observation of 2-methylsulfinylethanol are due to the slow non-enzymatic oxidation of 2-methylthioethanol after being formed by *E. coli* under oxic conditions (Fig. S4B). There is no 2-methylsulfinylethanol observed during anaerobic growth (16, 24). Ultimately, the 2-methylthioethanol produced by *E. coli* with the DHAP shunt serves primarily as a terminal product that can slowly oxidize during aerobic growth conditions to 2-methyl-sulfinylethanol.

## The *E. coli* DHAP shunt is not a detoxification pathway for SAM untilization byproducts

All *E. coli* possess the conserved Pfs enzyme, a multifunctional SAH, MTA, and 5dAdo nucleosidase (15). This functions to cleave the SAM utilization byproducts into adenine and the corresponding ribosyl species (Fig. 1A and B). In the absence of a functional *pfs* gene, accumulation of SAH, MTA, and 5dAdo competitively inhibits the activity of SAM-dependent enzymes, leading to slower cell growth (19, 41, 42). Historically for *E. coli*, the cleavage of SAH, MTA, and 5dAdo by the Pfs nucleosidase has been considered to be the sole and sufficient means for eliminating these inhibitory compounds, and MTR and 5dR generated by the cleavage of MTA and 5dAdo, respectively, are excreted as terminal byproducts in *E. coli* without the DHAP shunt (Fig. 1) (15). Recently, a possibly long-overlooked inhibitory role of MTR and 5dR was observed (17). In the insect pathogen *Bacillus thuringiensis*, the DHAP shunt kinase, isomerase, and aldolase were required for maximal

growth in glucose media if exogenously supplied 5dR was present. At concentrations of 1 mM 5dR, a ~25% reduction in growth rate was observed for DHAP shunt deletion strains compared to the wild type (17). This led to the hypothesis that the DHAP shunt's role is to detoxify internally produced or exogenously present MTR and 5dR, providing a potential growth advantage for strains possessing the DHAP shunt.

To assess the inhibitory effects of MTA, 5dAdo, MTR, and 5dR on the growth of *E. coli* with and without the DHAP shunt, and the physiological role of the conserved Pfs nucleosidase versus the DHAP shunt, we performed growth inhibition studies (Fig. 3A through C). In both the wild-type ATCC 25922 and K-12 strains, MTA and 5dAdo were inhibitory to growth at concentrations above 2 mM. When the *pfs* gene was inactivated in either strain, sensitivity to MTA and 5dAdo increased two- and fivefold, respectively (Fig. 3D). In stark contrast, when the DHAP shunt *mtnK, mtnA,* and *ald2* in ATCC 25922 were deleted (Fig. 3A and B; strain ΔK2), there was no statistically significant change in sensitivity to MTA and 5dAdo, indicating the DHAP shunt MtnK, MtnA, and Ald2 did not aid in managing the inhibitory effects of MTA and 5dAdo, and by extension MTR and 5dR. This establishes the essentiality of the Pfs nucleosidase in *E. coli* strains with and

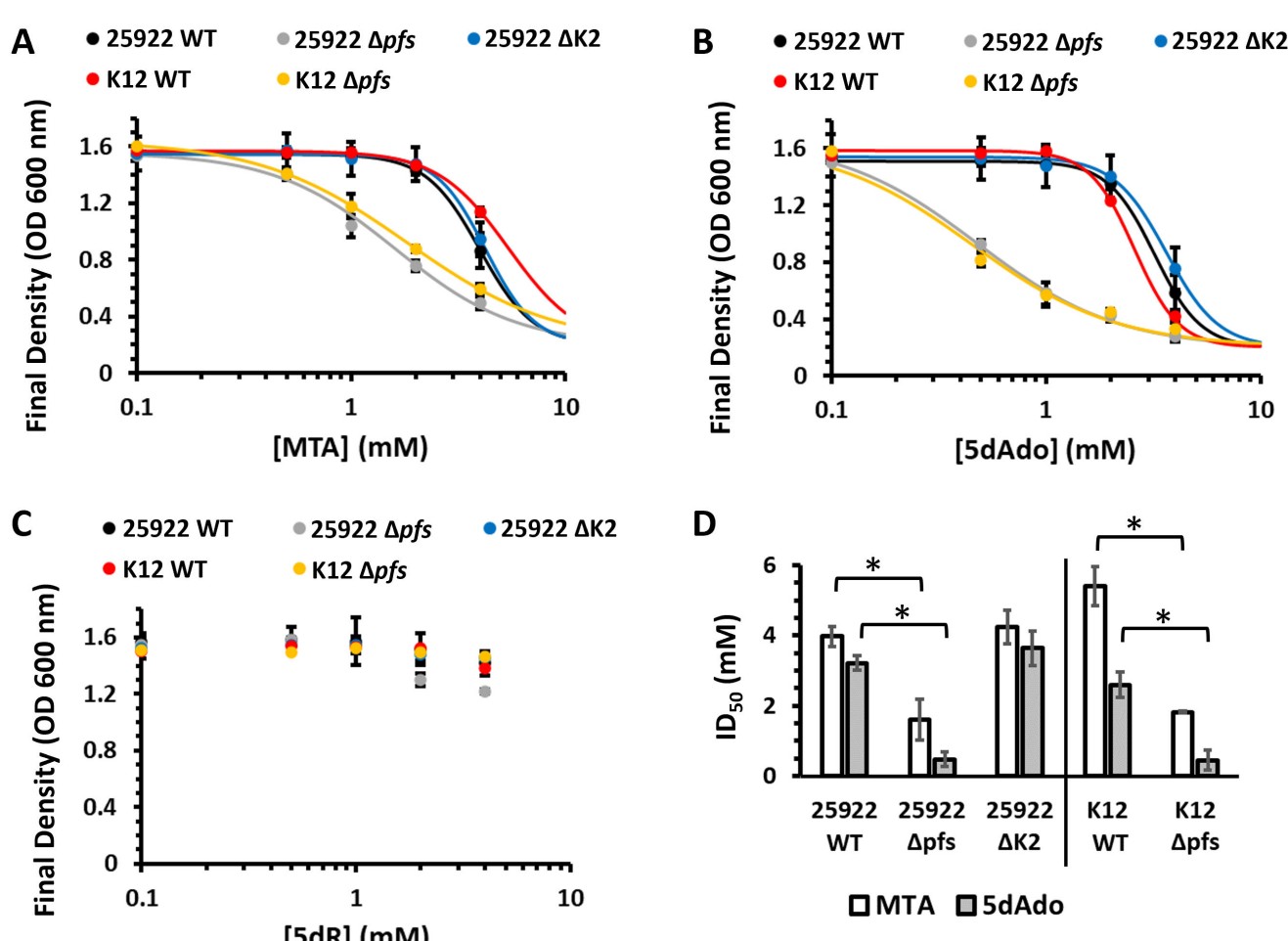

**FIG 3** Prevention of growth inhibition by SAM utilization byproducts requires Pfs but not the DHAP shunt. Final culture density after 18 hours of the ATCC 25922 wild-type, Δ*pfs*, and ΔK2 (Δ*mtnK* Δ*mtnA* Δ*ald2*) strains and the K-12 BW25113 wild-type and Δ*pfs* deletion strains grown aerobically with glucose and in the presence of the indicated concentration of either (A) 5′-methylthioadenosine, (B) 5′-deoxyadenosine, or (C) 5-deoxy-D-ribose. Averages and standard deviation error bars are for *n* = 3 independent replicates. Curves are the nonlinear least squares weighted fit to the Hill equation. (D) ID$_{50}$ for MTA and 5dAdo in ExPEC ATCC 25922 and commensal K-12 BW25113 strains. ID$_{50}$ values are from the fit curves in panels A and B to the Hill equation, and the error bars are the parameter 95% confidence interval of the weighted fit. Statistically significant difference, *P* < 0.05.

without the DHAP shunt to prevent the inhibitory buildup of SAM utilization byproducts consistent with previous reports (19, 41). We also directly quantified if 5dR was inhibitory. Even at a concentration of 4 mM supplied 5dR, there was no significant inhibitory effect on the growth of either ATCC 25922 or K-12 (Fig. 3C). Furthermore, deletion of the DHAP shunt by virtue of inactivating *mtnK, mtnA,* and *ald2* had no effect on ATCC 25922 sensitivity to 5dR (Fig. 3C). Thus, 5dR is not inhibitory in the millimolar range in which it can accumulate due to microbial metabolism (18).

## The *E. coli* DHAP shunt metabolizes SAM utilization byproducts for use as cellular growth substrates

As with *E. coli* lacking the DHAP shunt*,* many bacteria cannot utilize MTA and 5dAdo produced by SAM-dependent reactions beyond adenine salvage via the Pfs (MtnN) nucleosidase, resulting in the accumulation of the 5-deoxy-pentoses, MTR and 5dR (Fig. 1) (15, 18, 19, 25). Initially, we assessed the ability of *E. coli* strain ATCC 25922 to grow using 5dR compared to glucose during aerobic respiration. The strain grew aerobically with 5dR as a sole carbon source (Fig. 4A). When the DHAP shunt was deleted by the inactivation of the *mtnK*, *mtnA*, and *ald2* gene cluster (strain ΔK2), ATCC 25922 was completely incapable of growth with 5dR (Fig. 4A) but could still grow using glucose (Fig. S6). To verify that DHAP shunt inactivation was responsible for the loss of growth using 5dR, we reintroduced the DHAP shunt gene cluster in *trans* from a tetracycline-inducible plasmid, which fully restored aerobic growth with 5dR (Fig. 4A; pK2). This establishes that the DHAP shunt enables growth with 5dR as the sole C-source.

To further quantify the role of the DHAP shunt as a carbon acquisition pathway from 5′-deoxy-nucleosides, we grew ATCC 25922 and K-12 with MTA and 5dAdo as a sole C-source. Given that concentrations of these substrates above 1 mM are inhibitory (Fig. 3), we employed serial dilution plating using M9 agar plates supplemented with 1 mM glucose, MTA, or 5dAdo (Fig. 4B through G) instead of liquid cultures, which require ~5 mM substrate to quantify growth. The ATCC 25922 wild-type strain could utilize 5dAdo for growth similar to that of glucose (Fig. 4B and C). Growth with MTA was also observable but poor compared to glucose and 5dAdo (Fig. 4F). Consistent with the 5dR growth studies (Fig. 4A), deletion of the DHAP shunt resulted in minimal growth with 5dAdo and MTA, which was restored upon the expression of the DHAP shunt genes in *trans* from the tetracycline-inducible plasmid (Fig. 4D and E). Deletion of the Pfs nucleosidase in both strains resulted in poor growth on 1 mM glucose as well as 5dAdo and MTA, which was restored upon re-introduction of the *pfs* gene in *trans* (Fig. 4B through E). This underpins the importance of the nucleosidase in preventing the inhibitory accumulation of MTA and 5dAdo via intracellular SAM-dependent processes (Fig. 3) (19, 41). Thus, the DHAP shunt enables the metabolism of a diverse number of 5′-deoxy-nucleosides and 5-deoxy-pentoses for use as growth substrates.

Given that the efficiency by which an organism can convert a growth substrate into cell mass is dependent upon the nature of the growth substrate and how it can be metabolized, we assessed how efficiently *E. coli* could use 5dR versus other known growth substrates by measuring growth yield, $Y_{X/S}$ (gram of cell biomass per gram of substrate) under aerobic respiration, anaerobic respiration, and fermentation conditions (Fig. 5A). As expected for aerobic respiration, glycolytic and pentose phosphate substrates resulted in 0.5 g cells per gram of substrate, and pyruvate and lactate, which enter at the end of glycolysis before the TCA cycle, resulted in a yield of 0.3 g cells per gram of substrate (43, 44). The decreased efficiency for compounds of lower glycolysis (pyruvate, lactate, and DHAP) is due in part to an increased deleterious flux through the TCA cycle as wasted $CO_2$ (43) and/or due to being more oxidized than glucose, requiring a larger percentage of the substrate to be oxidized to $CO_2$ by the TCA cycle to generate enough reducing equivalents [NAD(P)H] to build cell biomass (45). The 5-deoxy-pentose sugar, 5dR, resulted in a yield of 0.3 g per gram of 5dR, just as for pyruvate. While growth with fucose and rhamnose 6-deoxy-hexoses was more efficient (0.4 g per gram of substrate), neither 5-deoxy-pentoses nor 6-deoxy-hexoses could be utilized as efficiently

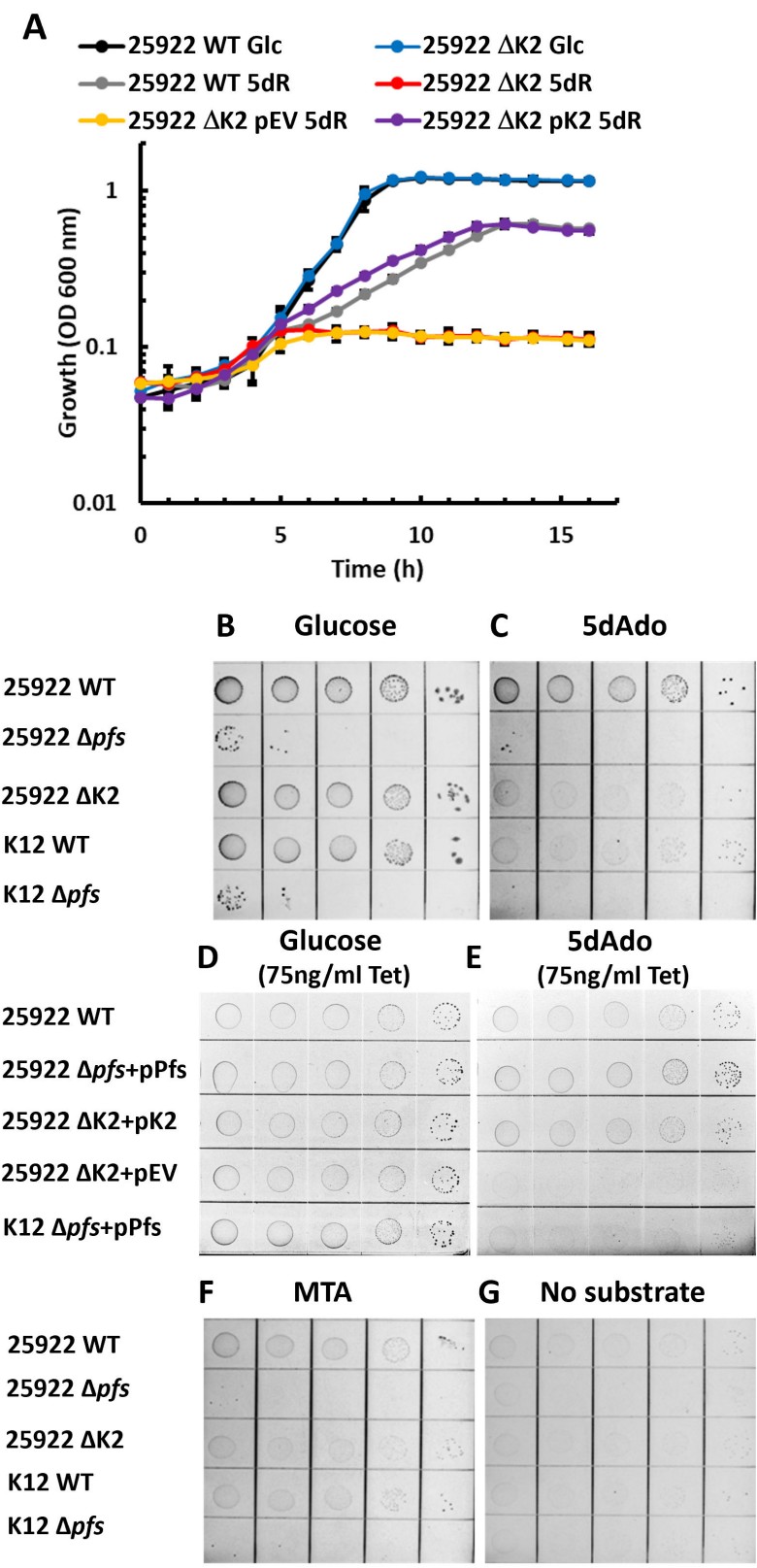

**FIG 4** Utilization of 5'-deoxy-nucleosides and 5-deoxy-pentoses for carbon and energy metabolism by *E. coli* with the DHAP shunt. (A) Growth of either wild-type ATCC 25922, ATCC 25922 ΔK2 (Δ*mtnK* Δ*mtnA* Δ*ald2*), ΔK2 + pEV (pTETTET empty vector), or ΔK2 + pK2 (pTETTET vector with DHAP shunt *mtnK, mtnA,*

**FIG 4** (Continued)

and *ald2*) on either 25 mM glucose (Glc) or 5 mM 5-deoxy-ᴅ-ribose as the carbon source and 50 ng/mL tetracycline. (B–G) Growth of ATCC 25922, ATCC 25922 Δ*pfs*, ATCC 25922 ΔK2, K-12 strain BW25113, or BW25113 Δ*pfs* on 1 mM of (B) glucose, (C) 5dAdo, (F) MTA, (G) or no carbon source. Coordinately, deletion strains were complemented with either an empty control plasmid (pEV), a plasmid containing the *mtnK, mtnA,* and *ald2* genes (pK2), or the *pfs* gene (pPfs) and grown on 1 mM of (D) glucose or (E) 5dAdo. A total of 75 ng/mL of tetracycline was added for gene expression from the plasmid tetracycline-inducible promoter (D and E).

as glucose. Thus, as for 6-deoxy-hexoses, the 5-deoxy-pentoses likely serve as secondary sugars when a preferred sugar like glucose is present (46, 47).

Under anaerobic conditions, *E. coli* ATCC 25922 could not grow fermentatively with 5dR like observed in other *E. coli* strains for lactate and ribose (Fig. 5A; Fig. S5B) (48, 49). Coordinately, it could grow fermentatively with glucose, pyruvate, and 6-deoxyhexoses as expected (Fig. 5A) (49). We verified that the DHAP shunt was active under anaerobic growth conditions through targeted metabolomics assays. Cells grown on glucose anaerobically and then fed with $^{3}$H-labeled 5dAdo showed conversion to 5dR and ultimately a mixture of acetate, acetaldehyde, and ethanol (Fig. 5B). This contrasts with aerobic growth in which only ethanol is formed (11). Given that ATCC 25922 could not ferment 5dR, we verified that 5dR could support anaerobic respiration with an appropriate electron acceptor. While nitrate did not support growth with 5dR as compared to glucose (Fig. S5A), TMAO supported anaerobic respiratory growth with 5dR as well as with the ribose and lactate substrates that also did not support fermentative growth (Fig. 5A). The anaerobic respiratory growth yield ($Y_{x/s}$) for glucose with TMAO was the same as previously reported for other strains of *E. coli* (0.2–0.3) (50). Notably, anaerobic respiratory growth yields for 5dR with TMAO were threefold lower than for glucose, and 1.5-fold lower than for pyruvate and lactate, revealing 5dR as a poor anaerobic respiratory substrate in comparison.

## The *E. coli* DHAP shunt supports aerobic and microaerobic respiratory growth with 5dR

ExPEC strains can inhabit a number of niches, including the intestine, urinary tract, and blood, which widely differ in oxygen and nutrient availability. While the intestinal tract is primarily anoxic, urine in healthy individuals typically has a dissolved $O_2$ concentration of about 4.2 ppm, which corresponds to ~0.065 atm (50 mm Hg) oxygen partial pressure (51). Oxygen concentration can vary with corresponding infection and urine microbial composition (52, 53). Given that the DHAP shunt supported the growth of ATCC 25922 with SAM utilization byproducts as carbon substrates under rigorously aerobic but not under fermentative conditions, this raised the question as to whether the DHAP shunt could support aerobic respiratory growth under oxygen tensions similar to those found in urine. Oxygen concentrations were varied from anaerobic to microaerobic, and cultures were monitored for growth using glucose or 5dR as the sole carbon source (Fig. 5C and D). The ATCC 25922 wild-type and DHAP shunt deletion strains (strain ΔK2) were able to grow using glucose across all oxygen concentrations tested (Fig. 5C). In stark contrast, ATCC 25922 wild-type strain could only grow aerobically using 5dR as a sole carbon source at oxygen tensions ≥ 0.004 atm (3 mm Hg). The observed growth with 5dR at the various oxygen tensions required the DHAP shunt, as no growth was observed with 5dR in the DHAP shunt deletion strain (Fig. 5D). Ultimately, the DHAP shunt appears to be physiologically relevant across the range of oxygen tensions found in typical extraintestinal environments for the use of SAM utilization byproducts as growth substrates.

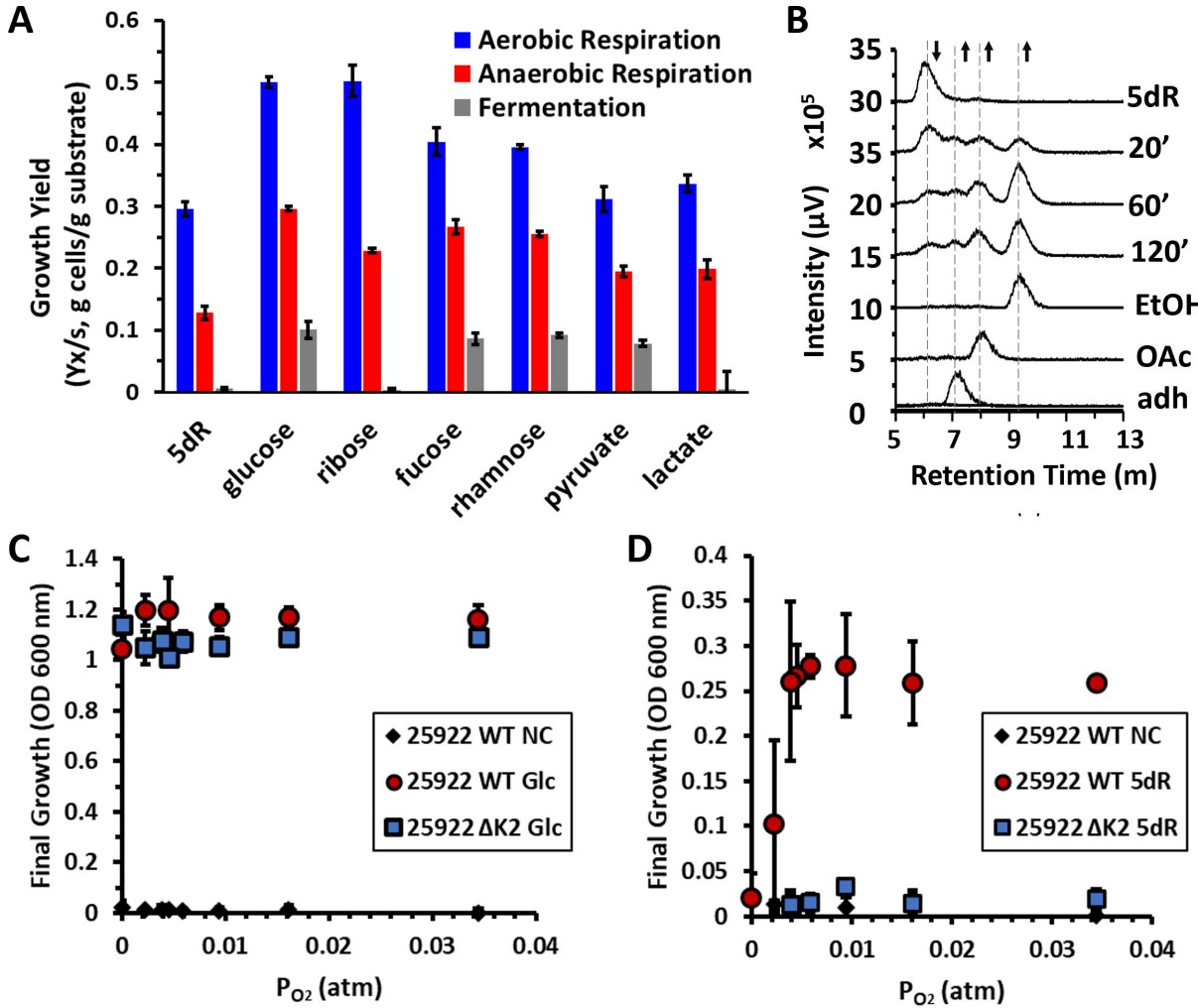

**FIG 5** The ExPEC DHAP shunt is an efficient carbon acquisition pathway for aerobic and microaerophilic conditions. (A) ATCC 25922 growth efficiency in the presence of 5 mM growth substrate measured as dry cell weight of cells generated per gram of substrate under aerobic respiration, anaerobic respiration with 100 mM TMAO, and under anaerobic fermentation growth conditions. EtOH, ethanol; OAc, acetate; adh, acetaldehyde. (B) Ion exclusion HPLC quantification of [³H]-labeled metabolites produced upon feeding *E. coli* ATCC 25922 with [³H-methyl]–5′-deoxyadenosine under anaerobic conditions. After feeding, metabolism was quenched by rapid freezing in liquid nitrogen at the indicated time in minutes. (C and D) ATCC 25922 wild-type and DHAP shunt deletion (ΔK2) strains were grown under varying oxygen concentrations with either (C) 25 mM glucose (Glc) or (D) 5 mM 5-deoxy-ᴅ-ribose (5dR) as the sole carbon source. In addition, experiments were performed with no carbon source (NC). In each experiment, 40 mM nitrate was also included, which was shown not to support anaerobic respiration (Fig. S5A). Final optical density measurements at 600 nm (OD₆₀₀) were taken after 24 hours of growth. Averages and standard deviation error bars in panels A, C, and D are for *n* = 3 independent replicates.

## Introduction of the DHAP shunt to *E. coli* K-12 enables growth with SAM utilization byproducts

The genes encoding *mtnK*, *mtnA*, and *ald2* of the DHAP shunt are not present in all *E. coli* and as such, *E. coli* lacking the DHAP shunt cannot grow using MTA, 5dAdo, or 5dR as the sole carbon source (Fig. 4). Given the DHAP shunt is located on the tRNA-*leuX* genomic island (11), we questioned if there are any other factors present in DHAP shunt-containing strains that are absent in *E. coli* lacking the DHAP shunt, which are required for growth using SAM utilization byproducts. We introduced the DHAP shunt gene cluster into *E. coli* K-12 and expressed the genes in *trans* from the same tetracycline-inducible plasmid used in ATCC 25922. The expression of *mtnK*, *mtnA*, and *ald2* enabled aerobic growth of K-12 with 5dR (Fig. 6). Thus, for aerobic growth and energy metabolism, *E. coli* can gain the ability to use SAM utilization byproducts through

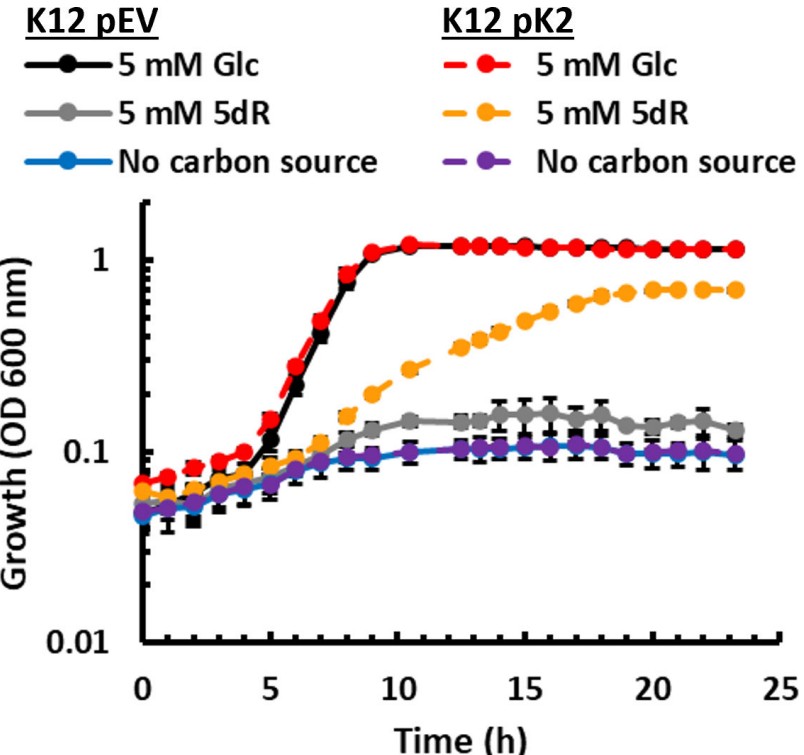

**FIG 6** Complementation of K-12 with DHAP shunt genes allows for growth using 5-deoxy-D-ribose as a carbon source. Growth of K-12 strain BW25113 complemented with pEV (pTETTET empty vector) and pK2 (pTETTET with DHAP shunt) using either 5 mM glucose (Glc) or 5 mM 5-deoxy-D-ribose as the carbon source, or no carbon source, and 50 ng/mL tetracycline. Average and standard deviation error bars are for $n = 3$ independent replicates.

the acquisition and expression of the DHAP shunt genes *mtnK*, *mtnA*, and *ald2* without any additional factors.

## DISCUSSION

SAM is a ubiquitous cofactor used by all organisms for methylations, polyamine synthesis, quorum sensing, and radical enzyme chemistry, which results in the formation of SAH, MTA, and 5dAdo (Fig. 1) (54). Subsequently, all organisms must eliminate these inhibitory compounds from the cell through further metabolism into less inhibitory species and/or excretion (15). While SAH is universally recycled by all organisms (except for some obligate endosymbionts) by a variation of the active methyl cycle (Fig. 1) (15), recent studies have revealed that a diversity of pathways exist for the detoxification and recycling of MTA (Fig. S1) (24, 35, 36, 55–57). These pathways can be bifunctional for both MTA and 5dAdo as in the case of the DHAP shunt (11, 17) or promiscuous for adenine as in the case of the universal methionine salvage pathway (58). Historically, the metabolism of MTA has been primarily considered to be a detoxification process to prevent inhibitory accumulation, and if recycled to methionine (Fig. S1), a sulfur salvage process with tie-ins to formate, isoprenoid, and ethylene synthesis (15, 16, 24, 35, 37). Analogously, the metabolism of 5dAdo has been considered to be a detoxification process (17, 19, 20), with tie-ins to 6-deoxy-5-ketofructose-1-phosphate production for amino acid biosynthesis in *Methanocaldococcus jannaschii* and 7-deoxysedoheptulose production as an anti-microbial secondary metabolite in *Synechococcus* sp. (59–61).

Unlike organisms in which the DHAP shunt was initially discovered to function as a methionine salvage pathway via the formation of 2-methylthioethanol as an intermediate (Fig. S1) (11, 16, 22–24), *E. coli* possessing the DHAP shunt does not appear to use

it for sulfur salvage and methionine synthesis from MTA (Fig. 2; Fig. S3D). Furthermore, in clear contrast to the insect pathogen, *Bacillus thuringiensis* (17), the DHAP shunt kinase (MtnK), isomerase (MtnA), and aldolase (Ald2) do not appear essential for the elimination of inhibitory byproducts of SAM utilization (Fig. 3). These DHAP shunt genes found in certain ExPEC strains and their deletion had no effect on the sensitivity of *E. coli* ATCC 25922 to MTA, 5dAdo, and 5dR. Only the Pfs nucleosidase, which is conserved across all *E. coli* for the hydrolysis of SAH, MTA, and 5dAdo, was absolutely required to prevent growth inhibition in the presence of these compounds. Thus, the conserved Pfs nucleosidase is the primary means for managing the inhibitory buildup of SAH, MTA, and 5dAdo produced by SAM-dependent enzymes. Coordinately, MTR and 5dR do not appear to need to be managed beyond export from the cell (18, 19).

The prime function of the DHAP shunt in *E. coli* that possess it is evidently for growth with 5′-deoxy-nucleoside and 5-deoxy-pentose sugars as C-sources (Fig. 4). The pathway does have a preference for 5dAdo and 5dR versus MTA and MTR as growth substrates, which is corroborated by our previous enzymatic analyses showing that the *E. coli* ATCC 25922 kinase (MtnK) had a 10-fold higher specificity for 5dR versus MTR ($k_{kat}/K_M = 6.3 \times 10^5$ versus $0.6 \times 10^5$ M$^{-1}$ s$^{-1}$) (11). Eukaryotic cells produce a diversity of 5′-deoxy-nucleosides as a byproduct of metabolism. These include MTA, 5′-methylthioinosine, 5dAdo, 5′-deoxyinosine, and 5′-deoxyxanthosine. In humans, these modified nucleosides each accumulate in the urine at concentrations of 0.1–10 μmol/mmol creatinine (1–300 μM based on average urine creatinine concentration range of 100–200 mg/dL) (62–65), which are within the physiological concentrations of metabolites often encountered and transported into the cell by *E. coli* (66). Notably for *E. coli*, the Pfs nucleosidase is active with each of these 5′-deoxy-nucleosides resulting in the formation of MTR or 5dR (62). Thus, the DHAP shunt is poised to be able to use carbon from a wide number of 5′-deoxy-nucleosides and the 5-deoxy-pentoses, MTR and 5dR, as substrates.

There are clear differences in the *E. coli* metabolism of 5′-deoxy-nucleoside and 5-deoxy-pentose substrates under aerobic versus anaerobic conditions in addition to the ability to perform respiration but not fermentation. Under aerobic conditions, the 5dR-derived acetaldehyde accumulates as ethanol as previously shown (11). This is because, during aerobic growth, *E. coli* cannot assimilate acetaldehyde and ethanol due to oxygen inactivation of the trifunctional CoA-acylating AdhE aldehyde/alcohol dehydrogenase for the metabolism of ethanol and acetaldehyde to acetyl-CoA. Rather, the acetaldehyde can only be converted to ethanol as a terminal product by alcohol dehydrogenase AdhA (49, 67). During anaerobic growth, the mixture of acetaldehyde, ethanol, and acetate (Fig. 5B) is due to AdhE activity, which reduces acetaldehyde to ethanol or oxidizes it to acetyl-CoA, depending on NAD+/NADH availability (49, 67). The inability of *E. coli* with the DHAP shunt to grow fermentatively with 5dR but grow via anaerobic respiration (Fig. 5A) even though the pathway is active anaerobically (Fig. 5B) supports the conclusion that *E. coli* mixed-acid fermentation cannot support sufficient ATP generation and/or redox balance with 5dR. Metabolism of 5dR results in pyruvate, acetaldehyde, one net ATP, and two reducing equivalents (2[H] as NADH) (Fig. 1). Based on carbon and oxidation balance for fermentation, the generated 2[H] limits the use of the 5dR-derived pyruvate through the *E. coli* mixed-acid fermentation phospho-trans-acetylase/acetate kinase (substrate-level phosphorylation) and formate-hydrogen lyase (electron transfer phosphorylation) pathways for further ATP generation, as compared to mixed-acid fermentation with pyruvate alone (Fig. 5A) (49). Furthermore, under anaerobic conditions, the mixed conversion of 5dR-derived acetaldehyde to ethanol and acetate (Fig. 5B) further supports the limitation of ATP production from 5dR by mixed-acid fermentation, as oxidation of acetaldehyde to acetate produces additional reducing equivalents that must be disposed.

Intriguingly, the respiratory conditions required by the DHAP shunt to support growth in *E. coli* that possess it potentially shed light on why the DHAP shunt appears predominantly in ExPEC strains (11). *E. coli* ATCC 25922 was incapable of growth on 5dR via fermentation or respiration with nitrate as an electron acceptor but was capable

of growth via aerobic respiration and anaerobic respiration with TMAO (Fig. 5). While the urine, blood, and mammary niches where ExPEC strains can inhabit are oxygenated at a sufficient level to enable the use of the DHAP shunt for aerobic respiratory growth with 5′-deoxy-nucleosides and 5-deoxy-pentoses as the sole C-source (Fig. 5) (51, 68), the large intestine is predominantly anoxic to microaerobic (13). For *E. coli*, mouse intestinal studies have shown that microaerobic respiration via cytochrome *bd* oxidase and anaerobic respiration with nitrate are essential for colonization in the gut. Conversely, alternate electron acceptors like DMSO, TMAO, or nitrite do not support *E. coli* colonization due to either low alternate electron acceptor abundance or the lack of functional terminal reductases (13). In support of the former, TMAO is primarily formed in the liver from gut microbiome-produced trimethylamine (14). As such, TMAO is readily found in both blood and urine but not found in significant quantities in the gut (69, 70). Thus, the ability to use TMAO and oxygen for respiratory growth with 5′-deoxy-nucleosides and 5-deoxy-pentoses via the DHAP shunt would be more relevant for ExPEC strains in their extraintestinal niches versus gut *E. coli*. Currently, whether the DHAP shunt in the ExPEC that possess it serves a growth fitness advantage remains to be determined. Regardless, given that the DHAP shunt is widespread across bacteria and in some pathogenic species of many genera, including *Bacillus* and *Clostridium* sp. (e.g., *Bacillus cereus*, *Bacillus anthracis*, *Clostridium botulinum*, and *Clostridium tetani*) (11), these findings that the DHAP shunt can function in carbon and energy metabolism for cell growth call for deeper investigation in both aerobic and anaerobic bacteria on its role beyond a 5′-deoxy-nucleoside and 5-deoxy-pentose detoxification pathway as initially proposed (17).

## ACKNOWLEDGMENTS

The authors thank the OSU Campus Chemical Instrument Center and Dr. Gong Wu for providing untargeted metabolomics support and use of the Thermo QE M UPLC MS/MS System.

These resources are supported in part by the NIH Award Number P30 CA016058. This work was funded by the NIH NIAID grant 1R01AI154456-01 (J.A.N. and F.R.T.).

K.A.H., J.T.G., J.A.W., and J.A.N. performed the growth and genetic analyses. K.A.H. prepared samples for the untargeted metabolomics and performed the targeted metabolomics. J.T.G. performed the metabolite quantification analyses. J.A.N. and F.R.T. designed the experiments and acquired funding. J.A.N. supervised the research. All authors discussed the research and contributed to writing the manuscript.

## AUTHOR AFFILIATION

[1]Department of Microbiology, The Ohio State University, Columbus, Ohio, USA

## PRESENT ADDRESS

John A. Wildenthal, Medical Scientist Training Program, Washington University in St. Louis School of Medicine, St. Louis, Missouri, USA

## AUTHOR ORCIDs

Katherine A. Huening ⓘ http://orcid.org/0009-0007-4706-1790
Joshua T. Groves ⓘ http://orcid.org/0009-0001-6673-5435
Justin A. North ⓘ http://orcid.org/0000-0002-4210-5463

## FUNDING

| Funder | Grant(s) | Author(s) |
| --- | --- | --- |
| HHS | National Institutes of Health (NIH) | RO1 AI154456 | F. Robert Tabita |

| Funder | Grant(s) | Author(s) |
|---|---|---|
| | | Justin A. North |

## AUTHOR CONTRIBUTIONS

Katherine A. Huening, Data curation, Formal analysis, Investigation, Methodology, Resources, Writing – original draft | Joshua T. Groves, Formal analysis, Investigation, Methodology, Writing – original draft | John A. Wildenthal, Formal analysis, Investigation, Resources, Writing – original draft | F. Robert Tabita, Conceptualization, Funding acquisition | Justin A. North, Conceptualization, Data curation, Formal analysis, Funding acquisition, Investigation, Methodology, Project administration, Supervision, Validation, Writing – original draft

## ADDITIONAL FILES

The following material is available online.

### Supplemental Material

**Supplemental material (Spectrum03086-23-S0001.docx).** Tables S1 and S2 and Figures S1 to S6.

### Open Peer Review

**PEER REVIEW HISTORY (review-history.pdf).** An accounting of the reviewer comments and feedback.

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
