## [Reviewer comments · Microbiology Spectrum]

Microbiology Spectrum

Escherichia coli possessing the Dihydroxyacetone Phosphate Shunt utilize 5'-deoxynucleosides for growth

Katherine Huening, Joshua Groves, John Wildenthal, F. Tabita, and Justin North

Corresponding Author(s): Justin North, The Ohio State University

Review Timeline:

Submission Date:	August 12, 2023
Editorial Decision:	September 25, 2023
Revision Received:	January 5, 2024
Editorial Decision:	January 15, 2024
Revision Received:	February 12, 2024
Accepted:	February 17, 2024

Editor: Silvia Cardona

Reviewer(s): The reviewers have opted to remain anonymous.

Transaction Report:

DOI: <https://doi.org/10.1128/spectrum.03086-23>

September 25, 2023

Dr. Justin Andrew North
The Ohio State University
Microbiology
423 Biological Sciences
484 W. 12th Avenue
Columbus, OH 43235

Re: Spectrum03086-23 (Extraintestinal Pathogenic E. coli utilize 5'-deoxynucleosides for growth via the Dihydroxyacetone Phosphate Shunt)

Dear Dr. Justin Andrew North:

Thank you for submitting your manuscript to Microbiology Spectrum. Your work has received favourable comments from reviewers as well as points that need further clarification. In particular, I agree with reviewers that the idea of SAM utilization by-products as a fitness trait in extra intestinal niches is highly speculative. Please pay special attention to the reviewers' suggestions regarding conclusions in anaerobic vs. aerobic conditions.

Link Not Available

Sincerely,

Silvia Cardona

Journals Department
Reviewer comments:

Reviewer #1 (Comments for the Author):

The authors did a thorough experimental research on the function of DHAP shunt in ExPEC. The experiments were well designed to address the question, all the figures were nicely made, and the topic raises important fundamental question.

Reviewer #2 (Comments for the Author):

In this study, the authors investigated the physiological role of the DHAP shunt in *E. coli* strains ATCC 25922 and BW25113. To learn more about the role of the DHAP shunt in *E. coli* strain ATCC 25922 regarding sulfur salvage from SAM utilization products. For this purpose, they tested whether *E. coli* ATCC25922 and different mutants lacking DHAP shunt-related genes grew with 5'-methylthioadenosine as sole sulfur source. This strain was unable to grow on MTA as sole sulfur source. Deletion of the DHAP shunt genes in strain ATCC 26922 did not result in impaired growth under sulfate-limiting conditions, suggesting that the DHAP shunt is not significantly involved in the salvage of MTA. To find out whether the DHAP shunt contributes to the detoxification of SAM byproducts, growth inhibition assays were performed in the presence of different 5'-deoxy-nucleosides. Whereas the deletion of the Pfs nucleosidase-encoding gene required for purine salvage increased the sensitivity to 5'-deoxy-nucleosides, the sensitivity was not affected when the DHAP shunt genes were deleted. Interestingly, the authors showed that the DHAP shunt was involved in providing carbon sources from SAM utilization byproducts under aerobic, but not under anaerobic growth conditions. Heterologous expression of the DHAP shunt genes *mtnK-mtnA-ald2* in trans resulted in growth of the *E. coli* K-12 derivative BW25113 on SAM utilization by-products as sole C-source. The authors interpret their findings as a fitness trait of ExPEC in nutrient-poor environments such as urine as the DHAP shunt may broaden the spectrum of compounds that could be used for carbon assimilation and energy metabolism in relevant environments. The observation that (i) *E. coli* ATCC 25922 uses the DHAP shunt as a means of carbon assimilation from external 5'-deoxy-nucleosides and 5-deoxy-pentose sugars for growth and (ii) that the DHAP shunt supports aerobic growth from 5'-deoxy-nucleosides and 5-deoxy-pentoses is in principle a very interesting observation that increases our knowledge about the metabolic versatility and potentially also about fitness traits of certain extraintestinal pathogenic *E. coli* variants. However, the extent to which this ability can be considered a fitness trait of ExPEC needs to be analysed in detail.

Points that need to be considered in more detail:

1) The wording that the *mtnK-mtnA-ald2* gene cluster is present in ExPEC and absent in IPEC and commensal is too simplistic and misleading. In fact, the comparative genome analysis summarized in Fig. S6 published in North et al. 2020 (DOI: 10.1111/mmi.14459) shows that the prevalence of this determinant varies between 14 % and 90 %, depending on the strain collection, with an average of 42% of all putative ExPEC isolates tested. As it is currently described in the manuscript, it is an oversimplification that is scientifically incorrect in this form implying that the DHAP shunt would represent a substantial fitness advantage for all ExPEC. However, this is not convincingly supported by the results which are based on the in vitro analysis of metabolic traits of one selected "ExPEC" isolate and one K-12 derivative.

The genomic basis of the current study (published in North et al. 2020) is too weak to make the statement that ExPEC, but not IPEC and commensals use the DHAP shunt truly meaningful. The prevalence on average of the *mtnKA-ald2* genes is not really high to support the idea that the encoded trait would be of general importance. Possibly it plays a context-dependent positive role for fitness of particular clones and lineages rather than in ExPEC in general?! The genome composition and molecular epidemiology of the different strain collections (tested in the publication North et al. 2020) should be considered with care regarding redundancy of clones. Furthermore, it would be important to systematically examine the *mtnK-mtnA-ald2* operon. Do the *mtn*-positive isolates have metabolic commonalities compared to the *mtn*-negative isolates, i.e. are certain pathways intact/functional and others not or absent? How conserved are the other SAM utilisation genes/pathways in these isolates? Similarly, a larger collection of fecal isolates from healthy individuals that do not carry a significant number of known virulence genes should be analyzed if the authors want to make relevant conclusions regarding "commensal" isolates.

2) Designating *E. coli* strain BW25113 as commensal is not really correct. Due to their long history as laboratory-adapted model strains, the members of the K-12 lineage (which has been repeatedly passaged and subjected to ionising radiation, ultraviolet light and mutagens, resulting in a number of genetic lesions and variants which have lost the F plasmid, bacteriophage λ and the ability to produce many surface-associated structures) are certainly non-pathogenic due to their genome content. BUT: they are also significantly less fit and less diverse and versatile in genome content than "real" commensals that have not been propagated in the lab for decades. This could be relevant in the context of the postulated positive fitness effect of the *mtn* gene cluster.

To what extent is *E. coli* ATCC 25922 an attenuated ExPEC (p. 5, l. 95)? Although this clinical isolate from 1946 carries several ExPEC virulence genes (P fimbriae, alpha-hemolysin, sat, *agn43*) and appears to be a good biofilm former, it is primarily used as a reference strain for antibiotic susceptibility testing. How does the fitness of this strain compare with that of well-established ExPEC model isolates? This reviewer believes that the study would be more convincing if, in addition to the interesting metabolic analyses presented for ATCC25922 and BW25113, the key messages of the study were confirmed in multiple relevant clinical or commensal faecal isolates. Otherwise, general statements on the role of the DHAP shunt in ExPEC and other pathotypes are not convincing.

3) Based on the in vitro results presented in the manuscript, would it not be appropriate to test the hypothesis that the DHAP shunt represents a fitness advantage for certain extraintestinal pathogenic *E. coli* under relevant in vivo conditions? For example, competition experiment could be performed in an experimental urinary tract infection model. In parallel, competition assays between a wild-type ExPEC and the corresponding *pfs* mutant or between *E. coli* K-12 and a *mtnK-mtnA-ald2*-expressing variant, for example, in human collected urine, would be convincing.

4) This is an interesting and principally well-designed study. It is, nevertheless, a great pity that in none of the experiments the authors made the effort to complement the effect of deleting the DHAP shunt or the *pfs* nucleosidase-encoding genes. A

properly conducted experiment should include the analysis of mutants and complemented mutants to document that the observed phenotype is indeed only due to the absence of the deleted genes. This gold standard should be implemented.

Minor points:

l. 431: "acetyl-COA" should read as "acetyl-CoA"

Reviewer #3 (Comments for the Author):

This manuscript addresses the role of the DHAP shunt in extraintestinal *E. coli*. The manuscript convincingly shows that the DHAP shunt can be used to assimilate deoxynucleosides at O₂ concentrations relevant to urine but that the shunt is not used for separate roles known for other bacteria. I consider the findings to be novel and interesting. The manuscript is easy to follow aside from some mildly disorientating figure panels described out of order. This manuscript was a rare pleasure to review.

Main concern:

1. I am glad the authors considered anaerobic respiration with nitrate but are the results from nitrate respiration conditions valid? I am concerned that there are no controls shown to verify anaerobic respiration with a non-fermentable carbon source by wildtype *E. coli* strains. It is also not clear what conditions starter cultures were grown in prior to growth with nitrate. What happens if cells are pre-grown on nitrate?

I suspect the substrates of interest are non-fermentable. The authors should could whether fermentation is possible in terms of avenues for electron balance and net ATP that could be generated by substrate level phosphorylation.

I was surprised that anaerobic respiration couldn't compensate for an inability to ferment the substrates of interest. Growth with nitrate can be complicated due to toxic nitrite generation. What about other electron acceptors for anaerobic respiration, like TMAO?

a. L367. data not shown?

b. L314 vs Fig S5. no nitrate mentioned in the legend. Does Fig S5 actually show fermentative conditions? Glucose is a fermentable substrate so even if nitrate is present, it might not be used.

c. FigS3 legend. nitrate present or not?

Minor comments:

2. L27. Oxic vs oxygenic

3. L77. 'essential' not necessary outside of the context of mammals

4. L93. No citation for previous report.

5. Perhaps it would be worth showing where methylthioethanol comes from in Fig 1 (adh activity)?

6. L307 - not sure why units are standardized to creatine. Wouldn't it be useful to give a typical mM concentration range of the compounds in interest so the reader can directly assess whether or not they present a significant C source in urine?

7. L316 - isn't growth of the dK2 strains also shown in Fig 4A? FigS6 is arguably excessive.

8. Fig 4B - why isn't the complemented strain also shown for these C-sources?

9. Fig 5A is actually a table. Why not make it a graph to allow for quick visual comparison

10. Fig 4F - nice closing experiment; a bit disorienting to have it appear as a subpanel in Fig 4

11. Fig 5C,D - very glad microaerobic conditions relevant to urine were addressed - nicely done

12. L189, L353, L422, elsewhere? Be more specific about which *E. coli* as these statements do not apply to K12.

13. L439, 449, strikingly x2

14. Unnecessary supplementary methods and results sections? I don't see anything about the supplementary methods, except

the promoter sequence, that would preclude inclusion in the methods. It doesn't look like methods count towards the word limit. There is also a surprise supplementary results section. If possible I'd prefer this be avoided. I suspect that most readers would prefer not to have supplementary materials if they can be avoided.

Staff Comments:

Preparing Revision Guidelines

Please return the manuscript within 60 days; if you cannot complete the modification within this time period, please contact me. If you do not wish to modify the manuscript and prefer to submit it to another journal, please notify me of your decision immediately so that the manuscript may be formally withdrawn from consideration by Microbiology Spectrum.

Response Letter

We are grateful to the Editor and Reviewers for their detailed review of the manuscript. We have addressed their comments, which we feel significantly increases the quality of the work, and have made overarching changes to the manuscript as follows:

- In general, we have shortened the introduction and discussion sections by removing the speculations on the use of the DHAP shunt as a fitness trait and by removing redundant overlap highlighting the general results of the paper and background on ExPEC strains. This has enabled relocation of the supplemental results and methods section back to the main article as requested by reviewer 3.
- Instead of referring to the *E. coli* strains used in the manuscript as commensal or ExPEC, we specifically refer to the strains as ATCC 25922 as an ExPEC representative with the DHAP shunt and K12 (BW25113) as a control strain without the DHAP shunt. This addresses both reviewer 2 and 3's comments regarding i) misleading conclusions that can be drawn by inferring all ExPEC behave like ATCC 25922 with the DHAP shunt and all IPEC and commensals behave like K12 (which can no longer be considered a commensal strain), and ii) preciseness on *E. coli* characteristics based on which *E. coli* strain is being used in the manuscript given ExPEC, IPEC, and commensals don't all share the same properties.

In addition, specific changes in response to the reviewers are detailed below.

Reviewer 1: We thank reviewer #1 for their positive assessment that the topic raises important fundamental question.

Reviewer #2:

The observation that (i) *E. coli* ATCC 25922 uses the DHAP shunt as a means of carbon assimilation from external 5'-deoxy-nucleosides and 5-deoxy-pentose sugars for growth and (ii) that the DHAP shunt supports aerobic growth from 5'-deoxy-nucleosides and 5-deoxy-pentoses is in principle a very interesting observation that increases our knowledge about the metabolic versatility and potentially also about fitness traits of certain extraintestinal pathogenic *E. coli* variants. However, the extent to which this ability can be considered a fitness trait of ExPEC needs to be analysed in detail.

Points that need to be considered in more detail:

1) The wording that the *mtnK-mtnA-ald2* gene cluster is present in ExPEC and absent in IPEC and commensal is too simplistic and misleading. In fact, the comparative genome analysis summarized in Fig. S6 published in North et al. 2020 (DOI: 10.1111/mmi.14459) shows that the prevalence of this determinant varies between 14 % and 90 %, depending on the strain collection, with an average of 42% of all putative ExPEC isolates tested. As it is currently described in the manuscript, it is an oversimplification that is scientifically incorrect in this form implying that the DHAP shunt would represent a substantial fitness advantage for all ExPEC.

However, this is not convincingly supported by the results which are based on the *in vitro* analysis of metabolic traits of one selected "ExPEC" isolate and one K-12 derivative. The genomic basis of the current study (published in North et al. 2020) is too weak to make the statement that ExPEC, but not IPEC and commensals use the DHAP shunt truly meaningful. The prevalence on average of the *mtnKA-ald2* genes is not really high to support the idea that the encoded trait would be of general importance. Possibly it plays a context-dependent positive role for fitness of particular clones and lineages rather than in ExPEC in general?! The genome composition and molecular epidemiology of the different strain collections (tested in the publication North et al. 2020) should be considered with care regarding redundancy of clones. Furthermore, it would be important to systematically examine the *mtnK-mtnA-ald2*-positive isolates for genomic commonalities and not limit the analysis to the gene neighbourhood flanking the *mtnKA-ald2* operon. Do the *mtn*-positive isolates have metabolic commonalities compared to the *mtn*-negative isolates, i.e. are certain pathways intact/functional and others not or absent? How conserved are the other SAM utilisation genes/pathways in these isolates? Similarly, a larger collection of fecal isolates from healthy individuals that do not carry a significant number of known virulence genes should be analyzed if the authors want to make relevant conclusions regarding "commensal" isolates.

We agree with Reviewer 2's assessment that as worded the wrong conclusion can be drawn that all ExPEC have the DHAP shunt. This in turn would lead to the wrong conclusion that the DHAP shunt has a meaningful relevance for all ExPEC. This is indeed not the case, and was not our intent. Rather, as the reviewer and the authors postulate, "Possibly it plays a context-dependent positive role for fitness of particular clones and lineages rather than in ExPEC in general." However, we are aware that definitive proof of the DHAP shunt in ExPEC as a fitness trait is not yet established and a focus of future *in vivo* work. Coordinately, our group is working on bioinformatic analyses of fecal sample databases for *E. coli* with and without the DHAP shunt by methods similar to those suggested by the reviewer to make truly meaningful statements about the frequency of the DHAP shunt in ExPEC versus commensal and IPEC stains. We feel these are beyond the scope of the current work due to their technically challenging and detailed nature, and the authors feel they are better suited for a future publication. Furthermore, as detailed in the North et al. 2020 Molecular Microbiology Paper, which we have also now referenced in the introduction, of the 1136 distinct IPEC isolates analyzed from the NCBI database, only 1 had the DHAP shunt genes. This is in contrast to the ExPEC strains, in which DHAP shunt genes were found in at least 6 different local genomic contexts, so even if there is high strain redundancy within collections as the reviewer appropriately cautions (which could bias in either direction), there is still a clear presence of the DHAP shunt in certain ExPEC strains over IPEC.

In response we have appropriately reworded the misleadingly dyadic treatment of ExPEC versus commensals with and without the DHAP shunt.

-We have slightly modified the title to "*Escherichia coli* possessing the Dihydroxyacetone Phosphate Shunt utilize 5'-deoxynucleosides for growth"

- Instead of referring to the pathways as the “ExPEC DHAP shunt”, we call it the “*E. coli* DHAP shunt” or “DHAP Shunt in *E. coli* that possess it”, which recognizes that the future planned bioinformatic analyses may find commensal or IPEC lineages of *E. coli* that contain the DHAP shunt.

- We have substantially revised the Importance and Discussion section to remove the assertion that the pathway would function in the context of infection. Rather the importance now focuses on the 5'-deoxynucleoside and 5-deoxypentose compounds that *E. coli* with the DHAP shunt can use during aerobic growth in oxic environments, but poorly use in anoxic environments.

- We now clearly state at the end of the discussion that it is not known whether the DHAP shunt serves as a fitness factor, or precisely why it appears at present to be predominantly found in some ExPEC strains, but put forth the idea as a testable hypothesis for future work.

2) Designating *E. coli* strain BW25113 as commensal is not really correct. Due to their long history as laboratory-adapted model strains, the members of the K-12 lineage (which has been repeatedly passaged and subjected to ionising radiation, ultraviolet light and mutagens, resulting in a number of genetic lesions and variants which have lost the F plasmid, bacteriophage λ and the ability to produce many surface-associated structures) are certainly non-pathogenic due to their genome content. BUT: they are also significantly less fit and less diverse and versatile in genome content than "real" commensals that have not been propagated in the lab for decades. This could be relevant in the context of the postulated positive fitness effect of the *mtn* gene cluster.

Reviewer 2 is correct and we appreciate them catching this oversight. We removed any instances of K12 being described as a commensal strain and appropriately described it as a control strain for those *E. coli* naturally lacking the DHAP shunt.

To what extent is *E. coli* ATCC 25922 an attenuated ExPEC (p. 5, l. 95)? Although this clinical isolate from 1946 carries several ExPEC virulence genes (P fimbriae, alpha-hemolysin, *sat*, *agn43*) and appears to be a good biofilm former, it is primarily used as a reference strain for antibiotic susceptibility testing. How does the fitness of this strain compare with that of well-established ExPEC model isolates? This reviewer believes that the study would be more convincing if, in addition to the interesting metabolic analyses presented for ATCC25922 and BW25113, the key messages of the study were confirmed in multiple relevant clinical or commensal faecal isolates. Otherwise, general statements on the role of the DHAP shunt in ExPEC and other pathotypes are not convincing.

This question was addressed by the works of Hof and Hacker in the late 1980's in a model of septicemia in mice. Established hemolytic uropathogenic strain 536 is of high virulence, leading to sepsis and death within 2-3 days after the injection of 10^3 bacterial cells. The non-hemolytic variant of 536 strain is avirulent and is unable to multiply in spite of agents to aid in protecting the cells from initial host immune defense after injection. The hemolytic *E. coli* strain ATCC

25922 is intermediate in virulence, and hence termed “attenuated”. The ATCC 25922 strain establishes a quantifiable blood infection and bacterial counts per liver increase steadily until death occurs five to seven days after the injection of 10^4 bacteria. In addition, ATCC 25922 can establish infection in mice in the heart, liver, spleen, lung, and kidney after peritoneal injection as an ExPEC strain. Specifically as a UPEC, there are no reports we are aware of for ATCC 25922 in a UTI model.

We have detailed this and added the corresponding references to the introduction section the manuscript:

- Hof, H., Christen, A., Hacker, J. 1986 Comparative therapeutic activities of ciprofloxacin, amoxicillin, ceftriaxone and co-trimoxazole in a new model of experimental infection with *Escherichia coli*. *Infection* 14:190-194
- Hof, H. and Fabrig, J., 1988. Comparative activities of norfloxacin and fleroxacin in experimental infections due to *Salmonella typhimurium* and *Escherichia coli*. *Journal of Antimicrobial Chemotherapy*, 22(Supplement_D), pp.123-127.
- Long, N., Deng, J., Qiu, M., Zhang, Y., Wang, Y., Guo, W., Dai, M. and Lin, L., 2022. Inflammatory and pathological changes in *Escherichia coli* infected mice. *Heliyon (Cell Press)*, 8(12).

Recent phylogenetic analysis using restriction marker and virulence gene identification shows that ATCC 25922 is closely related to UPEC strain CFT073 (B2 phylotype), and out of 9 characteristic ExPEC virulence genes (*cnf*, *papGII*, *papGIII*, *hlyC*, *sfa*, *fimH*, *usp*, *aer* and *fyuA*), both ATCC 25922 and CFT073 are only missing the cytotoxic necrotizing factor 1 (CNF) and PapGIII adhesin. We have included in the introduction that ATCC 25922 is a relative of CFT073.

- Schwan, W.R., Briska, A., Stahl, B., Wagner, T.K., Zentz, E., Henkhaus, J., Lovrich, S.D., Agger, W.A., Callister, S.M., DuChateau, B. and Dykes, C.W., 2010. Use of optical mapping to sort uropathogenic *Escherichia coli* strains into distinct subgroups. *Microbiology*, 156(Pt 7), p.2124.

We also agree with the reviewer that additional work in established ExPEC strains and clinical isolates with the DHAP shunt (e.g. ST131, NA114) would support claims regarding a specific benefit/function for ExPEC strains. To address this, as detailed for point 1 above, the manuscript now focuses on the role that the DHAP shunt serves in *E. coli* when it is present (i.e. enable growth using 5'-deoxynucleosides and 5-deoxyribose sugars) instead of focusing on the DHAP shunt as a putative ExPEC fitness factor. This is substantiated by the original complementation data for growth with 5-deoxyribose (Fig 4A and new Fig. 6), and as detailed below for point 4, is further substantiated by the new complementation experiments for growth on 5'-deoxyadenosine (Fig. 4B-G). While we agree with the reviewer that direct measurement of DHAP shunt function in clinical ExPEC isolates is important, the fact that the DHAP shunt from ATCC 25922 (a group 2B phylotype) can complement BW25113 (a long laboratory passaged and mutated group A phylotype as the reviewer points out) for growth with 5'-deoxyadenosine and 5'-deoxyribose is significant *raison d'être* alone as a carbon assimilation pathway in *E. coli*. While experiments are planned in our lab to directly test the functionality of the DHAP shunt in

ExPEC isolates, we feel with the current restructuring of the manuscript to remove assertions of the DHAP shunt as a fitness trait that such experiments are better suited for future work in conjunction with point 1 above.

3) Based on the in vitro results presented in the manuscript, would it not be appropriate to test the hypothesis that the DHAP shunt represents a fitness advantage for certain extraintestinal pathogenic *E. coli* under relevant in vivo conditions? For example, competition experiment could be performed in an experimental UTI model. In parallel, competition assays between a wild-type ExPEC and the corresponding *pfs* mutant or between *E. coli* K-12 and a *mtnK-mtnA-ald2*-expressing variant, for example, in human collected urine, would be convincing.

We agree with the reviewer that these are excellent experiments to test the hypothesis as to whether the DHAP shunt serves as a fitness factor. However, these are beyond the current allowed BSL regulated activities for our lab by the University Institutional Biosafety Committee and Institutional Review Board for use of human/animal materials. Indeed, they are a topic of future work after appropriate certifications, protocols, and instrumentations are in place. However, given the focus of the paper is now on the function of the DHAP shunt as a carbon assimilation pathway and not as a putative fitness trait, we feel these experiments would be better suited for future work with points 1 and 2 above.

4) This is an interesting and principally well-designed study. It is, nevertheless, a great pity that in none of the experiments the authors made the effort to complement the effect of deleting the DHAP shunt or the *pfs* nucleosidase-encoding genes. A properly conducted experiment should include the analysis of mutants and complemented mutants to document that the observed phenotype is indeed only due to the absence of the deleted genes. This gold standard should be implemented.

Respectfully, it appears that the reviewer may have overlooked the detailed complementation studies of Fig. 4A in which we deleted the ATCC25922 DHAP shunt specific *mtn* genes (Δ K2 strain), showed loss of growth with 5-deoxyribose, and then showed that an empty complementation plasmid (pEV) could not complement for growth but a plasmid with the *mtn* genes (pK2) complemented for growth on 5-deoxyribose in liquid culture. These studies furthermore show in liquid culture that when the DHAP shunt *mtn* genes are introduced to BW25113 (a K12 derivative), growth is enabled on 5-deoxyribose (original Fig. 4F, now Fig. 6 in response to reviewer 3's comments). These results were detailed in results section "DHAP shunt metabolizes SAM utilization byproducts for use as cellular growth substrates". Indeed, we followed the gold standard as outlined by the reviewer.

However, we see a point of confusion may have arisen in that we did not show complementation for growth with 5'-deoxyadenosine linked to the *pfs* gene (original Fig. 4B-E). As detailed in the introduction and the beginning of the results section "*The E. coli DHAP shunt is not a SAM utilization byproduct detoxification pathway*" the function of the *E. coli pfs* gene product, a tri-functional MTA/SAH/5dAdo nucleosidase has been thoroughly established through gene deletion and complementation by other researchers in the field in its role in

cleavage of MTA/SAH/5dAdo into adenine and a 5-deoxypentose sugars to prevent inhibitory buildup (Choi-Rhee 2005; Chadieux 2002, reviewed in Parveen and Cornell 2011). While these previous works preclude any need to complement the *pfs* strain in the inhibition studies (Fig. 3), it was an oversight of the authors (as also pointed out by reviewer 3) to not show complement of the *pfs* gene deletion strain for growth with 5'-deoxyadenosine (Fig. 4).

To address this, we have now included our direct measurements of ATCC 25922 showing on serial dilution plates that when the *pfs* or *mtn* genes are deleted and introduced with an empty vector there is no restoration of growth, but when the *pfs* or *mtn* genes are reintroduced via plasmid in the respective deletion strain, growth is enabled (new Fig. 4D-E). Furthermore, in Fig 4B-E we show deletion and restoration of the *pfs* gene in BW25113 confers no capability for growth with 5'-deoxyadenosine, but is able to grow when introduced with the *mtn* genes from the ATCC 25922 DHAP shunt.

Minor points

I. 431: "acetyl-COA" should read as "acetyl-CoA"
- this has been fixed.

Reviewer #3:

This manuscript addresses the role of the DHAP shunt in extraintestinal *E. coli*. The manuscript convincingly shows that the DHAP shunt can be used to assimilate deoxynucleosides at O2 concentrations relevant to urine but that the shunt is not used for separate roles known for other bacteria. I consider the findings to be novel and interesting. The manuscript is easy to follow aside from some mildly disorientating figure panels described out of order. This manuscript was a rare pleasure to review.

We are thankful to the reviewer for their enthusiasm and positive comments regarding the manuscript. In addition, we are very thankful for the reviewer regarding their insights on the potential difficulties in the use of nitrate for *E. coli* respiratory growth due to formation of inhibitory nitrite. In my years of working with *E. coli*, nitrate has always been considered preferred due to the chemiosmotic energy yield that can be gained from membrane vectoral proton transfer by nitrate reductase. But when a growth substrate was poor, the compounded effect of nitrite inhibition was not considered. Therefore we have included results with TMAO as suggested by the reviewer and detailed below.

Main concern:

1. I am glad the authors considered anaerobic respiration with nitrate but are the results from nitrate respiration conditions valid? I am concerned that there are no controls shown to verify anaerobic respiration with a non-fermentable carbon source by wildtype *E. coli* strains. It is also not clear what conditions starter cultures were grown in prior to growth with nitrate. What

happens if cells are pre-grown on nitrate? I was surprised that anaerobic respiration couldn't compensate for an inability to ferment the substrates of interest. Growth with nitrate can be complicated due to toxic nitrite generation. What about other electron acceptors for anaerobic respiration, like TMAO?

We thank the reviewer for raising this important point. Cells were pre-grown on glucose in the presence of nitrate before switching to 5-deoxyribose with nitrate in our initial tests for respiration. But more importantly, further work suggested by the reviewer with TMAO does enable anaerobic respiration. As a standard control we have now included aerobic respiration, anaerobic respiration, and fermentation studies with glucose, pyruvate, and lactate (as well as ribose, fucose, and rhamnose for comparison). These are shown in figure 5A which is now a graph instead of a table as suggested by the reviewer below. As expected, substrates that don't support fermentative growth for previously studied *E. coli* strains like lactate and ribose (Eggleston 1959 and Sawers 2004) do not serve as growth substrates for ATCC 25922. Then for each of these fermentable and non-fermentable substrates, including 5-deoxyribose, addition of TMAO as terminal electron acceptor enables a measurable growth yield. For typical substrates like glucose, growth yields of 0.29 +/- 0.01 g cells per g substrate were the same as previously reported for anaerobic respiration with TMAO (Ishimoto 1978). Notably, growth yield with 5-deoxyribose was only 0.12 +/- 0.02, which was even less than for pyruvate and lactate at 0.19 +/- 0.02. Thus, while the initial conclusion that 5-deoxyribose could not support anaerobic fermentation was incorrect, these results are still consistent with the model that 5-deoxyribose is a relatively poor substrate for anaerobic respiration.

We have updated the results section on growth with 5-deoxynucleosides and 5-deoxypentose sugars accordingly and have added the corresponding references below. In addition we have moved the discussion of substrate diversity produced by eukaryotes initial in the results section to discussion section where it is more appropriate given we are now focusing on these compounds as growth substrates via the DHAP shunt and not asserting the DHAP shunt is a fitness factor.

Eggleston, L. and H. Krebs, Permeability of *Escherichia coli* to ribose and ribose nucleotides. *Biochemical Journal*, 1959. 73(2): p. 264.

Sawers, R.G. and D.P. Clark, Fermentative pyruvate and acetyl-coenzyme A metabolism. *EcoSal plus*, 2004. 1(1).

Ishimoto, M. and O. Shimokawa, Reduction of trimethylamine N-oxide by *Escherichia coli* as anaerobic respiration. *Zeitschrift für allgemeine Mikrobiologie*, 1978. 18(3): p. 173-181.

2. I suspect the substrates of interest are non-fermentable. The authors should consider whether fermentation is possible in terms of avenues for electron balance and net ATP that could be generated by substrate level phosphorylation. We thank the reviewer for raising this important question. While we did not include it initially in the manuscript, we also considered if the

inability of 5-deoxyribose to support fermentative growth was logical based on ATP yields and electron balance. Given the initial manuscript was more focused on the DHAP shunt as a potential growth factor in ExPEC strains in oxic environments, we had chosen to omit discussion on fermentation. However, with the refocusing of the paper we agree with the reviewer that this is an important point to discuss.

The inability of *E. coli* with the DHAP shunt to grow fermentatively with 5dR but grow via anaerobic respiration (Fig. 5A) even though the pathway is active anaerobically (Fig. 5B) supports the conclusion that *E. coli* mixed fermentation cannot support sufficient ATP generation and/or redox balance with 5dR. Metabolism of 5dR results in pyruvate, acetaldehyde, one net ATP, and two reducing equivalents (2[H] as NADH) (Fig. 1). Based on carbon and oxidation balance for fermentation, the generated 2[H] limits the use of the 5dR-derived pyruvate through the *E. coli* mixed-acid fermentation phospho-transacetylase/acetate kinase (substrate level phosphorylation) and formate-hydrogen lyase (electron transfer phosphorylation) pathways for further ATP generation, as compared to mixed-acid fermentation with pyruvate alone (Fig. 5A) (Sawers 2004). Furthermore, under anaerobic conditions the mixed conversion of 5dR-derived acetaldehyde to ethanol and acetate (Fig. 5B) further supports the limitation of ATP production from 5dR by mixed acid fermentation, as oxidation of acetaldehyde to acetate further produces additional reducing equivalents that must be disposed.

We have now included this in the discussion

3a. L367. Data not shown? This is now included in Fig. 5 and Fig. S5B.

b. L314 vs Fig S5. No nitrate mentioned in the legend. Does Fig S5 actually show fermentative conditions? Glucose is a fermentable substrate so even if nitrate is present, it might not be used.

We thank the reviewer for pointing out this lack of clarity. Growth curves for both nitrate respiration (Fig. S5A) and fermentation (Fig. S5B) are now included and clearly labeled. In addition, a note in the figure legend has been added to indicate that the starter cultures were pre-grown on glucose under the same conditions.

c. FigS3 legend. Nitrate present or not? 40mM nitrate was present in all of these experiments, which has now been indicated in the figure caption. Given all of these experiments were performed with glucose, which *E. coli* can use for anaerobic respiratory growth with nitrate, the results are valid.

Minor comments:

2. L27. Oxic vs oxygenic The reviewer is correct that it should be oxic, which has been corrected

3. L77. 'essential' not necessary outside of the context of mammals. This has been removed

4. L93. No citation for previous report. This has been added (North, 2020, molecular microbiology) in the context of new lines 72-80 in response to reviewer 2. Line 93 has been deleted coordinately.

5. Perhaps it would be worth showing where methylthioethanol comes from in Fig 1 (adh activity)? Yes, this has now been added to figure 1.

6. L307 - not sure why units are standardized to creatine. Wouldn't it be useful to give a typical mM concentration range of the compounds in interest so the reader can directly assess whether or not they present a significant C source in urine? We thank the reviewer for this helpful suggestion. First we note that creatinine was accidentally misspelled as creatine in the original manuscript, which has been corrected. In the field of urine analysis normalizing compound amount to creatinine is commonly done due to creatinine and other metabolites being produced at a fairly constant daily rate irrespective of urine volume production rate. Thus normalizing to creatinine accounts for urine volume fluctuations. However, the reviewer is correct in that with respect to *E. coli* growth, the substrate concentration is important. We have now included the concentration range base on average creatinine ranges and have indicated that these concentrations of 1-300 μ M are similar to those ranges for other useful growth substrates encountered in the human environment.

- Peekhaus, N. and Conway, T., 1998. What's for dinner?: Entner-Doudoroff metabolism in *Escherichia coli*. *Journal of bacteriology*, 180(14), pp.3495-3502.

As part of restructuring the discussion to not assert that the DHAP shunt is a fitness train, we have moved the details on the diversity of 5'-deoxynucleosides and 5-deoxypentoses to the discussion where it is mor appropriate.

7. L316 - isn't growth of the dK2 strains also shown in Fig 4A? FigS6 is arguably excessive. While While Fig. S6 shows growth of the dK2 strain, this is specifically for glucose instead of 5dR as shown in 4A. Given the *pfs* gene deletion showed decreased growth on glucose, these additional controls were necessary to show that deletion of *mtnK/mtnA/ald2* to construct the dK2 strain did not affect growth on glucose, as indicated in the text "When the DHAP shunt was deleted by the inactivation of the *mtnK*, *mtnA*, and *ald2* gene cluster (strain Δ K2), ATCC 25922 was completely incapable of growth with 5dR (Fig. 4A), but could still grow using glucose (Suppl. Fig. S6)." Due to space considerations for showing the most pertinent data, these controls for glucose growth were not shown in Fig. 4A but rather relegated to supplement Fig. 6. On this point the authors feel that the data is best served as currently presented.

8. Fig 4B - why isn't the complemented strain also shown for these C-sources? We thank the reviewer for catching this. The complementation data is now shown, which shows that reintroduction of the DHAP shunt genes or the *pfs* gene restores growth with 5-deoxyadenosine.

9. Fig 5A is actually a table. Why not make it a graph to allow for quick visual comparison We thank the review rfor this helpful suggestion. The aerobic respiratory data along with the new anaerobic respiratory and fermentation data are now in bar graph format.

10. Fig 4F - nice closing experiment; a bit disorienting to have it appear as a subpanel in Fig 4 We thank the reviewer for this helpful suggestion, we have moved Fig. 4F to new Fig. 6.

11. Fig 5C,D - very glad microaerobic conditions relevant to urine were addressed - nicely done
Thank you

12. L189, L353, L422, elsewhere? Be more specific about which *E. coli* as these statements do not apply to K12. We thank the reviewer for pointing out this ambiguity. In conjunction with modifications made in response to reviewer 2, instead of referring to *E. coli* as commensal or ExPEC, we have specifically referred to the *E. coli* used by their strains (ATCC 25922 and K12), and in instances discussing the DHAP shunt include the distinction that these observations apply to those *E. coli* that possess it versus those that do not. For the *E. coli* variation of the DHAP shunt in those that possess it we now refer to it as the "*E. coli* DHAP shunt".

13. L439, 449, strikingly x2
This has been corrected.

14. Unnecessary supplementary methods and results sections? I don't see anything about the supplementary methods, except the promoter sequence, that would preclude inclusion in the methods. It doesn't look like methods count towards the word limit. There is also a surprise supplementary results section. If possible I'd prefer this be avoided. I suspect that most readers would prefer not to have supplementary materials if they can be avoided.

We agree with the reviewer that this should be avoided if possible. Initially, our longer introduction and discussion involving speculation of the role of the DHAP shunt as a fitness trait based on the poor fermentation and anaerobic respiration phenotype, and background on ExPEC strains meant that a supplemental results section was the best mechanism for detailing these data. However, removal of discussion on the DHAP shunt in ExPEC as a fitness factor and restructuring the paper to be less focused on overarching statements about ExPEC versus IPEC and commensals has shortened the main manuscript. In result, we have moved these methods and results sections back to the main paper.

Re: Spectrum03086-23R1 (Escherichia coli possessing the Dihydroxyacetone Phosphate Shunt utilize 5'-deoxynucleosides for growth)

Dear Dr. Justin Andrew North:

Thank you for reviewing your manuscript. While both reviewers were in agreement with the changes and improvements introduced, one reviewer has minor comments that should be addressed before the manuscript is ready for publication.

Please return the manuscript within 30 days; if you cannot complete the modification within this time period, please contact me.

Revision Guidelines

Sincerely,
Silvia Cardona
Editor
Microbiology Spectrum

Reviewer #2 (Comments for the Author):

Thank you for your careful consideration of the reviewer comments. By modifying the manuscript and omitting an explicit association of the dihydroxyacetone phosphate shunt with extraintestinal pathogenic E. coli, the data presented support the key messages of the manuscript.

Reviewer #3 (Comments for the Author):

I am satisfied with how the authors addressed reviewer comments aside from one lingering minor concern.

Main comment (Line numbers refer to the marked-up manuscript):

The growth data with TMAO weakens the argument that the DHAP shunt is associated with oxic environments. I am not overly concerned as there is plenty of novelty to the results overall and this argument was always going to be speculative. However, the authors could consider incorporating the TMAO data into the discussion with consideration of TMAO availability in relevant habitats. Personally, I wasn't convinced that growth with TMAO was relatively poor because (i) defining poor growth is somewhat subjective and (ii) any growth is strongly selective compared to not growth, so in the right environment, the availability of TMAO could select of the DHAP shunt.

Other comments:

L20. coil vs coli

L23. I think this sentence needs revision. The first part doesn't seem helpful, essentially stating 'Here we show the shunt?'

L23, and elsewhere. 'Carbon acquisition pathway' isn't intuitive to me. What about something like '... but rather enables growth on 5Ado and MTA as C sources.' If growth occurs when only one C source is provided then it is safe to say that C source was assimilated.

L410. Lactate misspelled

L545. Change to 'mixed-acid'

Dear Dr. Cardona and Reviewers,

We thank the Reviewers for their helpful suggestions to strengthen the paper and overcome a residual weakness. We have addressed the Reviewer's comments below and have updated the manuscript accordingly. We feel that focusing on where TMAO as an electron acceptor for growth with 5'-deoxynucleosides are expected to be available, versus considering growth with 5'-deoxynucleosides and TMAO to be a subjectively "poor" growth phenotype strengthens the overall manuscript.

Reviewer #3: Main comment (Line numbers refer to the marked-up manuscript):

The growth data with TMAO weakens the argument that the DHAP shunt is associated with oxic environments. I am not overly concerned as there is plenty of novelty to the results overall and this argument was always going to be speculative. However, the authors could consider incorporating the TMAO data into the discussion with consideration of TMAO availability in relevant habitats. Personally, I wasn't convinced that growth with TMAO was relatively poor because (i) defining poor growth is somewhat subjective and (ii) any growth is strongly selective compared to not growth, so in the right environment, the availability of TMAO could select of the DHAP shunt.

We thank the reviewer for pointing out this weakness and suggesting a discussion of TMAO availability. The reviewer is correct that even if anaerobic respiratory growth with TMAO is relatively poor using 5-nucleosides versus other carbon substrates, it could still have a positive selective pressure in the right environment. Given that *E. coli* ATCC 25922 could not use nitrate but could use TMAO for anaerobic respiration, previous work by Tyrell Conway and colleagues indicates that there would be little to no selective pressure in the gut environment for the DHAP shunt. In "Anaerobic Respiration of *Escherichia coli* in the Mouse Intestine" [Jones, *et al.*, mSphere. 2011. 79(10)], contributions of terminal reductases and terminal electron acceptors of *E. coli* for gut colonization were tested in a mouse model. Mutants lacking nitrate reductase or fumarate reductase had severe colonization defects. Conversely, mutants lacking DMSO and TMAO reductases (*dmsABC*, *torCAD*) had little to no defect in colonization. Together with earlier work [Jones *et al.* 2007. Infect. Immun. 75: 4891–4899] these results indicate that *E. coli* growth and colonization is not supported by respiration of DMSO, TMAO, or nitrite, "either because these electron acceptors are unavailable or because the terminal reductases are not functionally expressed". Thus, respiration of nitrate and fumarate plus fermentation, are likely the anaerobic processes that contribute to the colonization and growth of *E. coli* in the mouse intestine.

Coordinately, we would expect this to be true in the human intestine as well. Although the TMAO precursor, trimethylamine (TMA), is produced in the gut from microbial metabolic processes, its conversion to TMAO primarily occurs in the liver [Brown and Hazen, Nature Review Micro, 2018. 16(3): 181-181]. As such, TMAO is readily found in both blood and urine, but not found in significant quantities in the gut [Awaad *et al.*, Journal of Chromatography B. 2016. 1038: 12-18] [Hou *et al.*, Journal of Chromatography A. 2016. 1429: 207-217], indicating use of TMAO would be more relevant for ExPEC *E. coli* strains than commensal *E. coli*.

Thus, to remove the subjective "poorer growth" rationale for lack of presence of DHAP shunt-containing *E. coli* in the gut and to focus on environments where respiration with nitrate/TMAO/DMSO is relevant we have updated the abstract, importance, and *las* paragraph as follows:

We have changed L24 in the abstract to: The DHAP shunt in ATCC 25922 is active under oxic and anoxic conditions. Growth using 5-deoxy-ribose was observed during aerobic respiration and anaerobic respiration with TMAO, but not during fermentation nor respiration with nitrate. This suggests the DHAP shunt may only be relevant for Extraintestinal Pathogenic *E. coli* (ExPEC) lineages with the DHAP shunt that inhabit oxic or TMAO-rich extraintestinal environments.

We have change L43 in the importance to: This provides insight into the diversity of sugar compounds accessible by *E. coli* with the DHAP shunt and suggests that the DHAP shunt is primarily relevant in oxic or TMAO-rich extraintestinal environments.

We have altered the beginning of the last paragraph to: Intriguingly, the respiratory conditions required by the DHAP shunt to support growth in *E. coli* that possess it potentially sheds light on why the DHAP shunt appears predominantly in ExPEC strains [11]. *E. coli* ATCC 25922 was incapable of growth on 5dR via fermentation or respiration with nitrate as an electron acceptor but was capable of growth via aerobic respiration and anaerobic respiration with TMAO (Fig. 5). While the urine, blood, and mammary niches where ExPEC strains can inhabit are oxygenated at a sufficient level to enable use of the DHAP shunt for aerobic respiratory growth with 5'-deoxy-nucleosides and 5-deoxy-pentoses as the sole C-source (Fig. 5) [49, 66], the large intestine is predominantly anoxic to microaerobic [67]. For *E. coli*, mouse intestinal studies have shown that aerobic respiration via cytochrome bd oxidase and anaerobic respiration with nitrate are essential for colonization in the gut. Conversely, alternate electron acceptors like DMSO, TMAO, or nitrite do not support *E. coli* colonization due to either low alternate electron acceptor abundance or the lack of functional terminal reductases [67]. In support of the former, TMAO is primarily formed in the liver from gut microbiome-produced trimethylamine [68]. As such, TMAO is readily found in both blood and urine, but not found in significant quantities in the gut [69, 70]. Thus, the ability to use TMAO and oxygen for respiratory growth with 5'-deoxy-nucleosides and 5-deoxy-pentoses via the DHAP shunt would be more relevant for ExPEC strains in their extraintestinal niches versus gut *E. coli*.

Other comments:

L20. coil vs coli. This has been corrected; thank you

L23. I think this sentence needs revision. The first part doesn't seem helpful, essentially stating 'Here we show the shunt?'

In light of the below comment, we agree with the reviewer. This has been changed to "Rather, the DHAP shunt in *E. coli* ATCC 25922, and when introduced into *E. coli* K-12, enables the use of 5dAdo and MTA as a C-source for growth."

L23, and elsewhere. 'Carbon acquisition pathway' isn't intuitive to me. What about something like '... but rather enables growth on 5Ado and MTA as C sources.' If growth occurs when only one C source is provided then it is safe to say that C source was assimilated.

We thank the reviewer for noting this. The phrase "Carbon Acquisition" is a recent development from the marine and soil microbiology fields to distinguish between pathways/substrates for heterotrophic

growth (Carbon Acquisition) and pathways/substrates for autotrophic growth which is defined as “Carbon Assimilation”. For example: Muñoz-Marín, M.C., Gómez-Baena, G., López-Lozano, A. et al. Mixotrophy in marine picocyanobacteria: use of organic compounds by Prochlorococcus and Synechococcus. ISME J 14, 1065–1073 (2020). <https://doi.org/10.1038/s41396-020-0603-9>.

This is meant to avoid confusion, but is not yet widely accepted terminology. During the review we realized in our original manuscript that we had used the term “carbon assimilation” to indicate that the 5′-deoxynucleoside/5-deoxyribose substrate was incorporated into cell biomass, consistent with historical practice in the microbial physiology field (e.g. glucose assimilation).

In light of this, we agree the statement on L23 “functions as a carbon acquisition pathway for 5dAdo and MTA as growth substrates” may be confusing and is in fact redundant. This has been modified as detailed for the comment above.

-L29 “carbon acquisition and energy metabolism” has been changed to “carbon and energy metabolism”

-L97 “Here we report that *E. coli* ATCC 25922 uses the DHAP shunt as a means of carbon acquisition from externally acquired 5′-deoxy-nucleosides and 5-deoxy-pentose sugars for growth...” has been changed to “Here we report that *E. coli* ATCC 25922 uses the DHAP for growth with externally acquired 5′-deoxy-nucleosides and 5-deoxy-pentose sugars as a C-source...”

-L333 has been changed to “This establishes that the DHAP shunt enables growth with 5dR as the sole C-source”

-L448 has been changed to: “The prime function of the DHAP shunt in *E. coli* that possess it is evidently for growth with 5′-deoxy-nucleoside and 5-deoxy-pentose sugars as C-sources”

-L492 has been changed to: “The lower oxygen tensions found in extraintestinal environments are still sufficient to enable use of 5′-deoxy-nucleosides and 5-deoxy-pentoses as C-sources for growth via the DHAP shunt (Fig. 5)”

-L498 has been changed to: “these findings that the DHAP shunt can function in carbon and energy metabolism for cell growth call for deeper investigation”

L410. Lactate misspelled. This has been corrected; thank you

L545. Change to 'mixed-acid'. This has been corrected; thank you

Re: Spectrum03086-23R2 (Escherichia coli possessing the Dihydroxyacetone Phosphate Shunt utilize 5'-deoxynucleosides for growth)

Dear Dr. Justin Andrew North:

Your manuscript has been accepted, and I am forwarding it to the ASM production staff for publication. Your paper will first be checked to make sure all elements meet the technical requirements. ASM staff will contact you if anything needs to be revised before copyediting and production can begin. Otherwise, you will be notified when your proofs are ready to be viewed.

Sincerely,
Silvia Cardona
Editor
Microbiology Spectrum

Reviewer #3 (Comments for the Author):

The authors have done an excellent job addressing my comments. The thoughtful discussion they provided regarding their TMAO data really puts it in perspective. I sincerely appreciate the authors' patience with my 2nd review and the careful consideration given to my comments.

1. L98 - word missing: '...uses the DHAP for...' vs '...uses the DHAP shunt for...'

2. Nothing needs to be addressed for this comment, indeed the authors have already addressed it, but the review given in the rebuttal as an example of the uses of carbon assimilation vs acquisition, does not make this distinction between the two terms (<https://doi.org/10.1038/s41396-020-0603-9>). Rather, the review describes 'assimilation' of organic compounds:

'Light-stimulated amino acid assimilation reported in those areas might be attributed to both genera of cyanobacteria...'

'Two pathways have been designated in cyanobacteria for glucose assimilation...'

'Why does the assimilation of organic molecules represent a worthy investment?'

Personally, I would be against using 'acquisition' to signify assimilation of organic carbon sources (e.g., carbon could be acquired for energy transformation alone), but at the very least, such distinctions should be clearly defined, whenever used, to avoid confusion.